# Jellyfish genomes reveal distinct homeobox gene clusters and conservation of small RNA processing

Wenyan Nong [1,13], Jianquan Cao[1,13], Yiqian Li[1,13], Zhe Qu[1,13], Jin Sun[2], Thomas Swale[3], Ho Yin Yip[1],
Pei Yuan Qian [2], Jian-Wen Qiu [4], Hoi Shan Kwan[5], William Bendena [6], Stephen Tobe [7],
Ting Fung Chan [8], Kevin Y. Yip [9], Ka Hou Chu [10], Sai Ming Ngai[8], Karl Yk Tsim[11],
Peter W. H. Holland [12✉] & Jerome H. L. Hui [1✉]

The phylum Cnidaria represents a close outgroup to Bilateria and includes familiar animals including sea anemones, corals, hydroids, and jellyfish. Here we report genome sequencing and assembly for true jellyfish *Sanderia malayensis* and *Rhopilema esculentum*. The homeobox gene clusters are characterised by interdigitation of Hox, NK, and Hox-like genes revealing an alternate pathway of ANTP class gene dispersal and an intact three gene ParaHox cluster. The mitochondrial genomes are linear but, unlike in *Hydra*, we do not detect nuclear copies, suggesting that linear plastid genomes are not necessarily prone to integration. Genes for sesquiterpenoid hormone production, typical for arthropods, are also now found in cnidarians. Somatic and germline cells both express piwi-interacting RNAs in jellyfish revealing a conserved cnidarian feature, and evidence for tissue-specific microRNA arm switching as found in Bilateria is detected. Jellyfish genomes reveal a mosaic of conserved and divergent genomic characters evolved from a shared ancestral genetic architecture.

[1] School of Life Sciences, Simon F.S. Li Marine Science Laboratory, State Key Laboratory of Agrobiotechnology, The Chinese University of Hong Kong, Shatin, Hong Kong. [2] Department of Ocean Science, Division of Life Science and Hong Kong Branch of the Southern Marine Science and Engineering Guangdong Laboratory, The Hong Kong University of Science and Technology, Clear Water Bay, Hong Kong. [3] Dovetail Genomics, Scotts Valley, CA, USA. [4] Department of Biology, Hong Kong Baptist University, Kowloon Tong, Hong Kong. [5] School of Life Sciences, The Chinese University of Hong Kong, Shatin, Hong Kong. [6] Department of Biology, Queen's University, Kingston, ON, Canada. [7] Department of Cell and Systems Biology, University of Toronto, Toronto, ON, Canada. [8] School of Life Sciences, State Key Laboratory of Agrobiotechnology, The Chinese University of Hong Kong, Shatin, Hong Kong. [9] Department of Computer Science and Engineering, The Chinese University of Hong Kong, Shatin, Hong Kong. [10] School of Life Sciences, Simon F.S. Li Marine Laboratory, The Chinese University of Hong Kong, Shatin, Hong Kong. [11] Division of Life Science, Hong Kong University of Science and Technology, Clear Water Bay, Hong Kong. [12] Department of Zoology, University of Oxford, Oxford OX1 3SZ, UK. [13] These authors contributed equally: Wenyan Nong, Jianquan Cao, Yiqian Li, Zhe Qu ✉email: peter.holland@zoo.ox.ac.uk; jeromehui@cuhk.edu.hk

Bilaterians comprise over 99% of extant animal species. Comparing genome sequences between diverse bilaterian animals, including insects, nematodes, annelids, amphioxus and vertebrates, allows insight into the genes and genome organisation of their long-extinct last common ancestor, the urbilaterian[1–8]. For example, comparisons indicate the urbilaterian had a Hox gene cluster, an NK homeobox gene cluster and a ParaHox gene cluster, although these have been broken in some lineages[9–14]. Numerous individual genes can confidently be deduced to have been present in the urbilaterian[15], as can aspects of post-transcriptional regulation such as mechanisms for differential use of arms from a microRNA duplex[16,17]. To understand how bilaterian characters evolved, outgroups are necessary. The closest sister group to the Bilateria is either Cnidaria[18,19] or a Cnidaria plus Placozoa clade[20].

In recent years, the genomes of several cnidarian species have been published with varying levels of assembly contiguity. These include the anthozoans Nematostella vectensis, Aiptasia strain CC7 and Acropora digitifera[21–23], hydrozoans Hydra magnipapillata and Clytia hemisphaerica[24,25], myxozoans Kudoa iwatai and Myxobolus cerebralis[26], cubozoan Morbakka virulenta and the scyphozoans Aurelia and Nemopilema nomurai[27–29]. These data have given important insights into gene family evolution (for example[30,31]), but our understanding of animal genome evolution is still somewhat constrained by restricted taxonomic sampling and the limited contiguity of several genomes.

Jellyfish is the common name for the free-swimming form of gelatinous animals with bells and tentacles, especially the medusa phase of cnidarians, although the term is occasionally extended to ctenophores. Within cnidarians, the scyphozoans are sometimes referred to as 'true jellyfish', and like other cnidarians their body is constructed from two germ layers and their tentacles are armed with nematocysts with venom for capturing prey and/or defence against predators. Scyphozoans play significant ecological roles from surface waters to the deep sea, as an important part of the oceanic food chain, and they are found in every major ocean in the world. Scyphozoan jellyfish in coastal seas interact with humans in several ways. Thousands of swimmers are stung with varying degrees of severity every year. In addition, when their living conditions are favourable, scyphozoans can form swarms (jellyfish blooms), which can damage fishing apparatus or clog the cooling systems of power stations. Several species in the order Rhizostomae have been adopted as a food source in some regions and are farmed in aquaculture systems.

Here, we present two high-quality de novo reference genomes for Amuska jellyfish Sanderia malayensis and edible jellyfish Rhopilema esculentum. Unique and conserved genomic features, and aspects of post-transcriptional gene regulation are revealed. These genomic resources expand the gene repertoire of true jellyfishes or scyphozoans, and provide insights into the understanding of evolutionary pathways of both bilaterians and cnidarians.

## Results

**High-quality genomes of two jellyfish.** Genomic DNA was extracted from single individuals of two species of true jellyfish (Scyphozoa), Sanderia malayensis (fam. Pelagiidae, Fig. 1a) and Rhopilema esculentum (fam. Rhizostomatida, Fig. 1b), and sequenced with Illumina short-read, 10x Genomics linked-read, and PacBio long-read sequencing platforms (Supplementary Tables 1–2). Hi-C libraries were also constructed for both jellyfish and sequenced on the Illumina platform (Supplementary Fig. 1). We compared several assembly approaches that integrate different types of sequencing data, and selected the best approach for each assembly: using self-corrected PacBio reads for S. malayensis

followed by scaffolding with Hi-C data, and a hybrid approach using Illumina and PacBio reads for R. esculentum followed by merger of haplotypes and scaffolding with Hi-C data. The S. malayensis genome assembly is 184 Mb with a scaffold N50 of 4.6 Mb spanning 970 scaffolds with 26,914 predicted protein coding genes (Table 1). The high-physical contiguity is matched by high completeness, with 90.6% complete BUSCO genes (metazoa_odb9 dataset run in genome mode) (Table 1). The R. esculentum genome is 256 Mb with a scaffold N50 of 12.9 Mb and 87.1% BUSCO completeness (metazoa_odb9 dataset run in genome mode) with 18,923 predicted protein coding genes (Table 1). The R. esculentum genome assembly reaches close to chromosomal scale with ~94.72% of the sequences contained on 21 pseudomolecules (Supplementary Table 3).

Phylogenomic analysis places the two sequenced species together with other jellyfishes (including two Aurelia and the Nemopilema nomurai) in the class Scyphozoa, which is sister group to Cubozoa. These two clades, together with two Hydrozoa species, were clustered in Medusozoa (Fig. 1d).

Despite S. malayensis having the smallest cnidarian genome reported to date (Table 1), it contains a similar number of predicted genes (27,365) to other cnidarian genomes (ranging from 21,862 to 38,007) (Supplementary Tables 8–10). Concomitant with this, the average size of S. malayensis predicted protein coding genes (including UTRs deduced from transcriptome reads) is smaller than in other cnidarians analysed (~4.5 kb per gene). In addition, the mean size of introns is the smallest of all published cnidarian genomes (381 bp, Supplementary Table 10). In total, the length of DNA sequence contributing to coding genes in the S. malayensis genome is small (~123 Mb) compared to other cnidarian genomes (Supplementary Table 10). The only (high quality) cnidarian genome known with less DNA sequence contributing to genes is the sea anemone Nematostella vectensis genome (~114 Mb); however, in the latter this comprises only ~32% of the genome in comparison to the ~50% in S. malayensis. Thus, small exons and small introns are the major factors contributing to the small genome size of S. malayensis.

**Jellyfish homeobox gene clusters.** There is a wealth of comparative data concerning diversification and organisation of homeobox genes in animal genomes; these genes are of interest due to their roles in body patterning and their usefulness as markers of large-scale genomic changes in evolution. Within the ANTP class of homeobox genes, Hox, ParaHox and NK genes are arranged in gene clusters in some bilaterians, and are inferred to have split apart from an ancestral 'mega-homeobox cluster' before divergence of major bilaterian lineages[9–14]. Their primary roles in bilaterians may have been to pattern the anteroposterior axis of the nervous system (Hox) and gut (ParaHox), and functional subdivision of mesoderm (NK), but whatever the initial roles, selective pressures have maintained their gene linkages while many other ANTP class genes dispersed[32,33]. Cnidarians also possess a large diversity of ANTP class genes including, at least in some species, a small Hox cluster[21,34,35] and ParaHox gene pair[36] or triplet[28], but longer range homeobox gene organisation is unclear.

In the jellyfish S. malayensis, we identified 101 homeobox genes including 40 ANTP class, and similar numbers in R. esculentum (98/42; Supplementary Tables 14–20). Gene assignments are based on gene trees and synteny (Supplementary Figs. 3–9 and Supplementary Table 17). The scaffold size achieved allowed gene cluster arrangement to be determined in jellyfish (Fig. 2b). In S. malayensis, we find a Hox gene cluster including orthologues of anthozoan Hox genes and linked to Evx (scaffold 466), comparable but not identical to the arrangement in N. vectensis

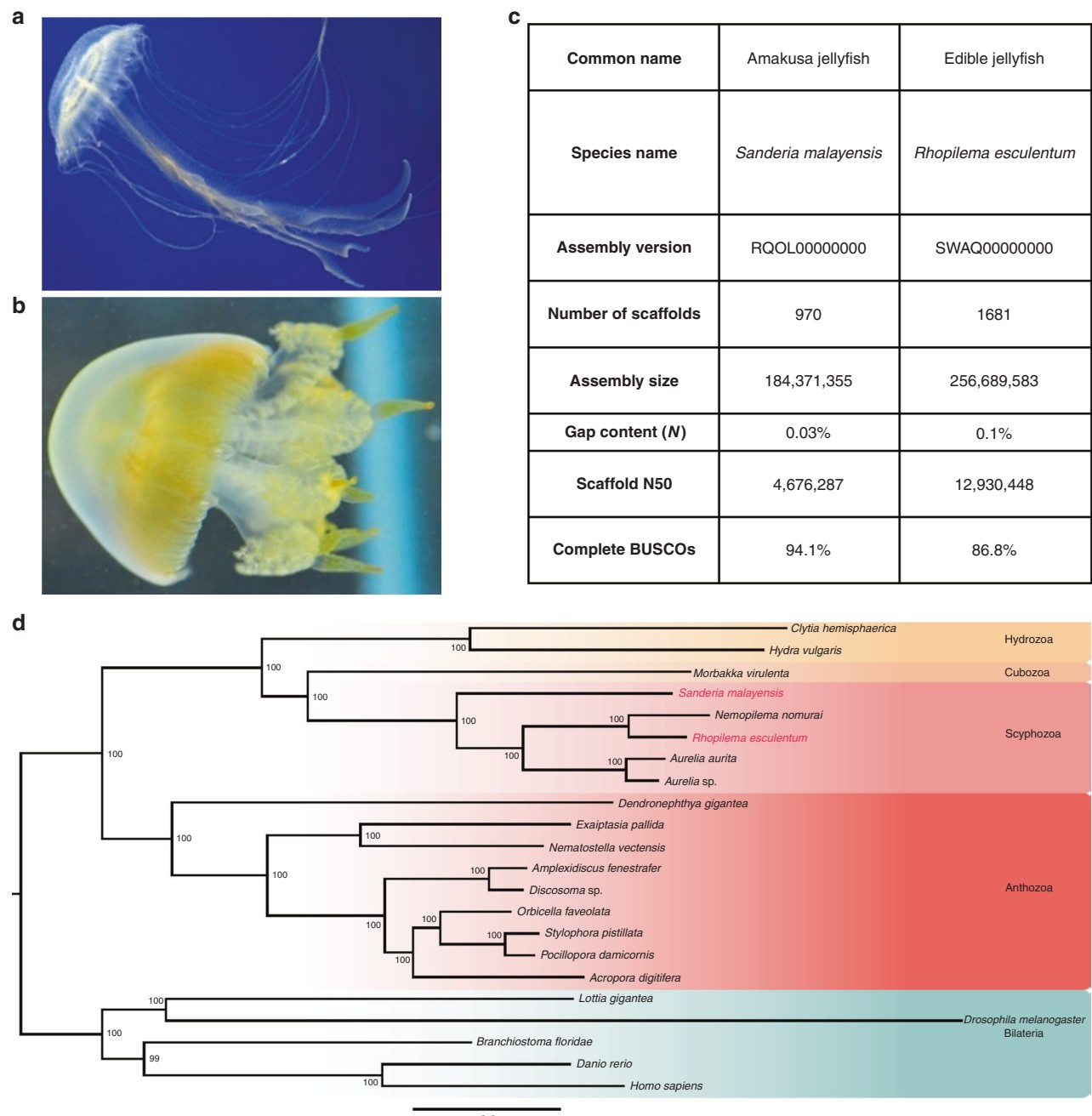

**Fig. 1 The two jellyfish models used in this study and their phylogenetic positions. a** Amuska jellyfish *Sanderia malayensis*; **b** Edible jellyfish *Rhopilema esculentum*; **c** Genome assembly quality. **d** Phylogenomic tree showing the positions of the two jellyfish (highlighted in red).

and *A. digitifera*[34,35]. Orthology to cnidarian and bilaterian Hox gene regions was confirmed by syntenic analysis using nearby genes (Fig. 2c and Supplementary Fig. 8). Interestingly, this scaffold also includes putative Dlx genes, mirroring the Hox-Evx-Dlx linkages of vertebrates and amphioxus[37–40]. In contrast to bilaterians, we find two other genomic regions containing putative Hox genes clustered with NK genes such as Emx, NK1, NK5 and Hhex (scaffolds 531 and 772). The Hox-Evx-Dlx linkage and the interdigitation of Hox and NK genes is also found in the *R. esculentum* genome (Fig. 2b).

A ParaHox gene cluster containing three homeobox genes was also identified in both jellyfish species (Fig. 2b and Supplementary Fig. 7), as also described in moon jellyfish *Aurelia*[28], and distinct from the single ParaHox gene or a two-gene cluster reported for

other Cnidaria[34–36]. Our analyses suggest the cluster includes likely orthologues of Pdx and Gsx, with the third gene being either Cdx or an independent duplication. Orthology to cnidarian and bilaterian ParaHox gene regions were also confirmed by syntenic analysis using nearby genes (Fig. 2c and Supplementary Fig. 8). Analyses of available transcriptome and genome data from other Cnidaria indicate that the third ParaHox gene is widespread amongst Medusozoa (including Staurozoa, Cubozoa, Scyphozoa and some Hydrozoa; Fig. 2b and Supplementary Table 20).

**Linear mitochondrial genomes**. Mitochondrial DNA (mtDNA) can insert into nuclear genomes, which can result in non-functional, gradually degrading, nuclear copies (NUMTs).

**Table 1 Comparison of cnidarian genome assembly quality.**

| Common name | Coral | Sea anemone | Sea anemone | Hydroid | Hydroid | Box jellyfish | Moon jellyfish | Moon jellyfish | Amuska jellyfish | Edible jellyfish |
|---|---|---|---|---|---|---|---|---|---|---|
| Species name | Acropora digitifera | Exaiptasia pallida | Nematostella vectensis | Hydra vulgaris | Clytia hemisphaerica | Morbakka virulenta | Aurelia (California) | Aurelia (Baltic sea) | Sanderia malayensis | Rhopilema esculentum |
| Assembly version | GCA_000222465.2 | GCA_001417965.1 | GCA_000209225.1 | GCA_000004095.1 | / | RDPX00000000 | | REGM00000000 | RQOL00000000 | SWAQ00000000 |
| Number of scaffolds | 2421 | 4312 | 10,804 | 20,916 | 7644 | 4538 | 25,454 | 2710 | 970 | 1681 |
| Assembly size | 447,497,157 | 256,132,296 | 356,613,585 | 852,170,992 | 445,209,699 | 951,575,644 | 757,170,055 | 376,952,359 | 184,371,355 | 256,689,583 |
| Gap content (N) | 15.24% | 17.69% | 16.61% | 7.82% | 16.63% | 11.87% | 12.84% | 6.63% | 0.03% | 0.10% |
| Contig N50 | 10,915 | 14,401 | 19,835 | 10,112 | 3,800 | 30,845 | 20,000 | 33,962 | 576,835 | 207,270 |
| Scaffold N50 | 483,559 | 442,145 | 472,558 | 96,317 | 366,311 | 2,173,999 | 121,658 | 1,042,981 | 4,676,287 | 12,930,448 |
| Complete BUSCOs | 74.6% | 88.7% | 89.1% | 81.5% | 86.0% | 81.5% | 86.0% | 79.8% | 90.6% | 87.1% |
| References | Shinzato et al.[23] | Baumgarten et al.[22] | Putnam et al.[21] | Chapman et al.[24] | Leclère et al.[25] | Khalturin et al.[28] | Gold et al.[27] | Khalturin et al.[28] | This study | This study |

The presence of NUMTs has been evaluated in several animal taxa, but is of particular interest in Cnidaria because of the unusual mitochondrial DNA in some lineages. Unlike the bilaterians, which have circular mtDNA, the hydroid *H. vulgaris* has two linear mtDNA molecules while the anthozoans *N. vectensis* and *A. digitifera* have circular mtDNA. As *H. vulgaris* has a far higher number of NUMTs than *N. vectensis* and *A. digitifera*, it has been suggested that linear mtDNA genomes may be especially prone to nuclear integration (e.g., ref. [41]).

Linear mitochondrial genomes were assembled in both *S. malayensis* and *R. esculentum* (Fig. 3), as expected for scyphozoans[42–45] (Supplementary Tables 11–13). We did not identify NUMTs in the nuclear genome assemblies of either jellyfish using sequence similarity searches. Since non-functional NUMTs should accumulate mutations, absence of NUMTs was further tested by mapping the Illumina raw reads to the assembled mitochondrial genome sequences. This approach did not detect any sequence variation, also consistent with lack of NUMTs in both jellyfish (Fig. 3).

**Cnidarian sesquiterpenoid pathway.** Many cnidarians undergo dramatic metamorphosis or developmental transitions, including budding in hydrozoans and strobilation in jellyfish. To date, very little is known about factors that regulate cnidarian life cycles[46]. In a recent study, genes encoding the neuropeptides eclosion hormone and bursicon, both formerly thought specific to insects, with eclosion hormone involved in ecdysone-regulated timing of moulting and bursicon involved in insect wing expansion during adult emergence, were found in cnidarian genomes[47]. This raises the possibility that other hormonal systems controlling insect metamorphosis could also be conserved in cnidarians.

The hormonal control of insect metamorphosis involves changing interaction between two hormonal systems: ecdysone for control of cuticle moulting and sesquiterpenoids (juvenile) hormones implicated in post-embryonic growth and differentiation[48]. We have focused on the putatively interacting sesquiterpenoid pathway, especially since in insects, sesquiterpenoids control the transition between developmental stages and are essential for reproduction[48–50]. Derived from an acetate precursor through the mevalonate pathway, farnesyl units are converted to cholesterol and steroid hormones in vertebrates, or into sesquiterpenoid hormones such as juvenile hormone in insects[51,52] (Fig. 4a). The biosynthetic pathway is uncharacterised in non-bilaterians.

In the cnidarian genomes investigated here, genes in the sesquiterpenoid biosynthetic pathway that were identified include protein farnesyl transferase (FNT), Ste 24 endopeptidase (ZMPSTE24), prenyl protein peptidase (RCE1), isoprenylcysteine carboxymethyl transferase (ICMT), prenylcysteine oxidase (PCYOXIL) and aldehyde dehydrogenase (ALDH) (Fig. 4b, c; Supplementary Fig. 13 and Supplementary Table 24). These enzymes could control the production of the sesquiterpenoid farnesoic acid (FA). FA is a biologically active stimulant of arthropod vitellogenesis[53], and it has been thought FA is restricted to the protostome lineage (Fig. 4b, c and Supplementary Fig. 13). The role of cnidarian FA in either reproduction or morphogenesis has yet to be determined. Previously, sesquiterpenoid methyl farnesoate has been found a non-arthropod bilaterian (annelid *Platynereis dumerilii*)[54]. Our findings show that genes for sesquiterpenoid hormone production, typical for arthropods, are also present in cnidarians.

**Ubiquitous piRNA in somatic cell, and microRNA arm switching.** Small RNAs are important regulators of gene activity in animals. Two of the major classes of small RNAs are

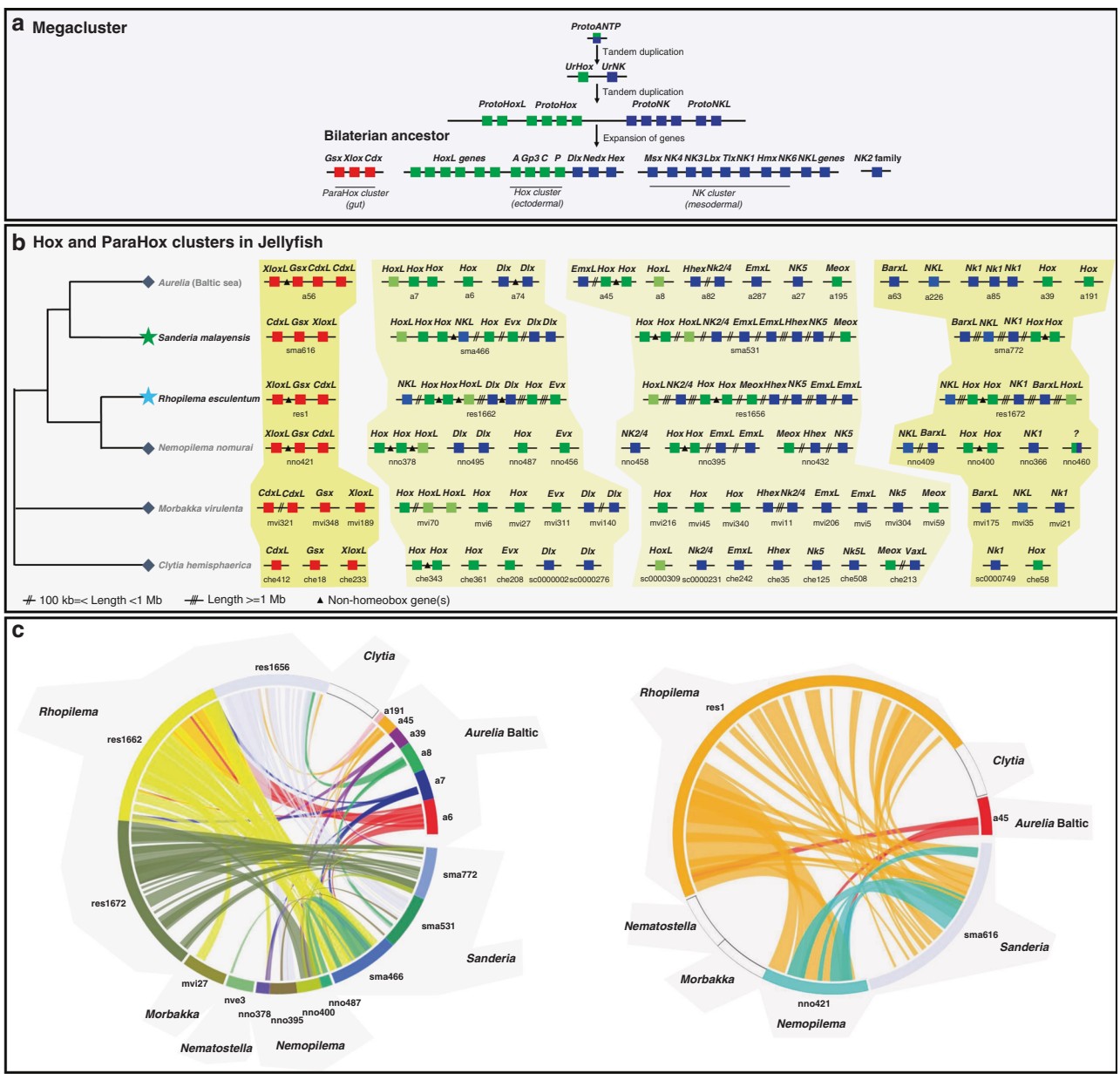

**Fig. 2 Homeobox genomic organisation. a** Schematic diagram showing origin of bilaterian homeobox gene clusters from a hypothesised ANTP class megacluster, with ParaHox cluster, Hox cluster, NK cluster, and NK2 genes located on separate chromosomes; **b** Schematic summary of ANTP-class homeobox gene arrangement in the jellyfish genomes; question mark denotes divergent ANTP-class homeobox genes; double slash denotes genomic distance >100 kb and <1 Mb; triple slash denotes genomic distance over 1 Mb; triangle denotes intervening non-homeobox gene and is only used when the distance <100 kb. A three gene ParaHox cluster is present in *S. malayensis*, and the interdigitation of Hox and NK cluster genes is recovered in *Rhopilema esculentum* and *S. malayensis*; **c** Syntenic relationships between scaffolds containing Hox and ParaHox genes in cnidarians supporting gene assignments.

microRNAs (21–23 nt) implicated in post-transcriptional gene regulation and piRNAs (>27 nt) primarily involved in suppression of transposable element activity[55]. Both are thought to play major roles in animal evolution, with microRNAs canalising development through suppression of transcriptional noise thereby facilitating the strength of natural selection[56–59] and piRNAs ensuring that mobile DNA is kept in check in the germline[60]. Although >30 microRNAs are conserved between divergent bilaterians, only the miR-100 family has been found to be shared between cnidarians and bilaterians[16,56,61].

We sequenced small RNAs from different tissues of the two jellyfish plus the moon jellyfish *Aurelia aurita* (Supplementary Tables 4–6), and checked authenticity by mapping to genome sequence and testing for predicted hairpin sequences. 71, 65,

and 149 putative microRNAs were annotated in *A. aurita*, *R. esculentum*, and *S. malayensis*, respectively; of these, 22, 41, and 125 have high confidence fulfilling all criteria in MirGeneDB (Supplementary Data 1–4). As with other cnidarians, the majority of confidently assigned microRNAs were species-specific, with only two—miR-2022 and miR-2030—shared between jellyfish and the anthozoan *N. vectensis* (Fig. 5a; Supplementary Fig. 11 and Supplementary Data 1–3). We also note that miR-100 seems to have been lost in the medusozoan lineages (Fig. 5a), and a total of six microRNAs are conserved across jellyfish genomes only (Supplementary Table 22).

In the biogenesis of microRNAs, after the formation of pre-microRNA duplexes, both 5p and 3p arms can potentially generate functional mature microRNAs[62]. In bilaterians, the

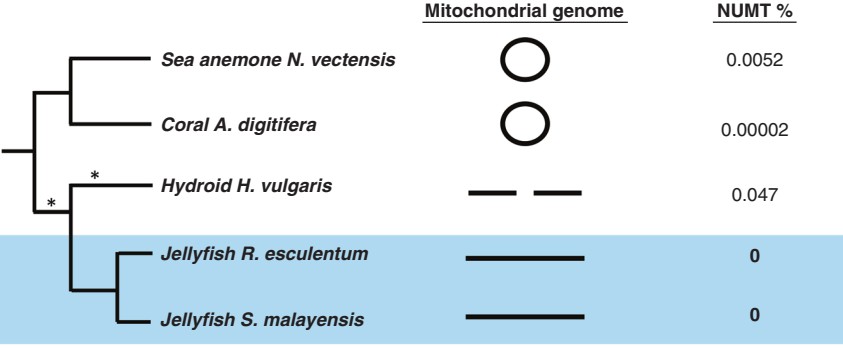

**Fig. 3 Mitochondrial genomes in cnidarians.** Diagram relating mitochondrial genome conformation to presence of nuclear copies of mitochondrial DNA (NUMTs) in cnidarian genomes. The situation in jellyfish genomes implies that linear plastid genomes are not necessarily prone to nuclear integration.

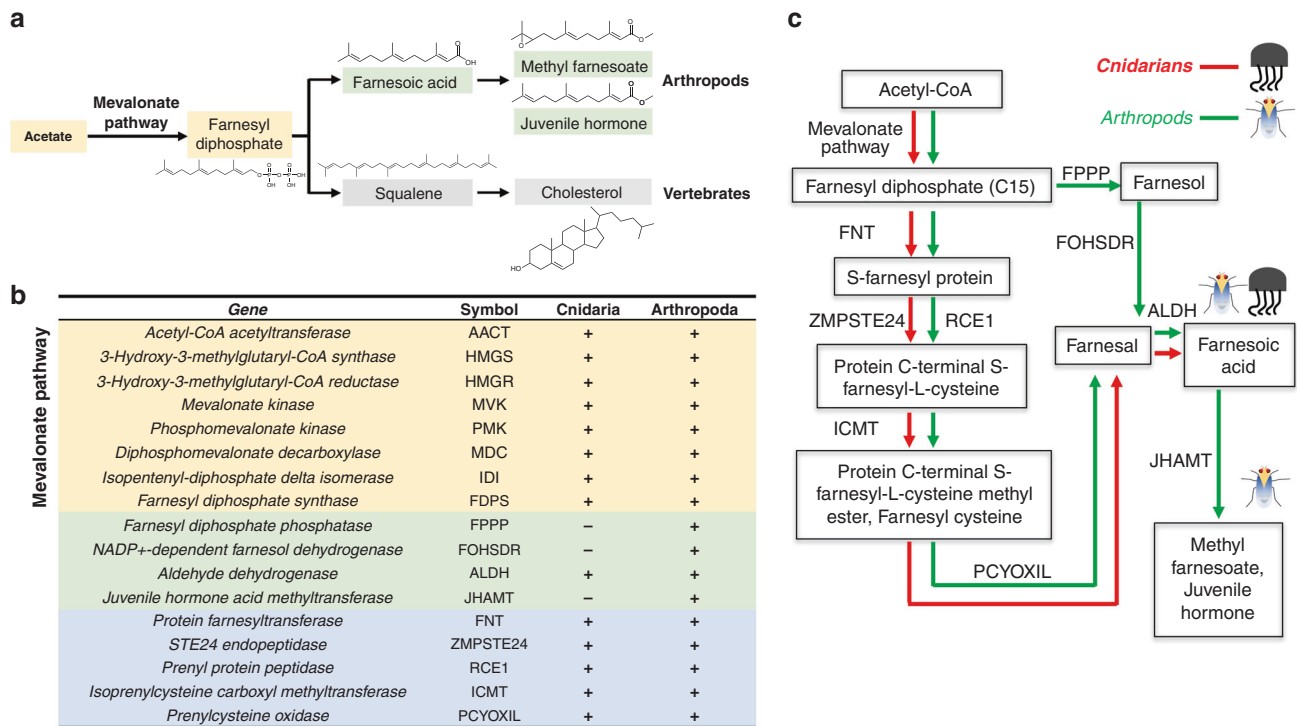

**Fig. 4 Sesquiterpenoids in cnidarians. a** Comparison of terpenoid and cholesterol biosynthesis in arthropods and vertebrates; **b** Comparison of gene homologues involved in the terpenoid backbone biosynthetic pathway and the hormone biosynthetic pathway in Cnidaria and Arthropoda, plus and minus denote the presence and absence of that gene in respective phyla; **c** Schematic diagram summarising terpenoid backbone biosynthesis and hormone production in Cnidaria and Arthropoda.

usage of the two arms can be modulated and swapped in dominance between different tissues, developmental stages or species; since 5p and 3p arms have different targeting properties, this 'arm switching' affords an additional level of regulatory and evolutionary flexibility[16,17,63–65]. We detected a case of micro-RNA arm switching in jellyfish. In *S. malayensis*, the microRNA ScYm1zk_729_32283 has 3p dominant arm usage in the rhopalia and 5p dominant arm usage in tentacles (Fig. 5d; Supplementary Table 23 and Supplementary Data 4). This is the first case of microRNA arm switching in non-bilaterians, suggesting the underlying mechanism was established before the cnidarian-bilaterian common ancestor.

We found that both microRNAs and piRNAs are expressed in germ cells (gonad) and somatic cells of jellyfish (Fig. 5b and Supplementary Fig. 12), as are the genes encoding argonaute and

PIWI proteins that associate with the small RNAs (two *Ago-like* genes, and two *PIWI* genes; Fig. 5c and Supplementary Fig. 10). Expression of the PIWI machinery is considered important for protection of the germline in bilaterians, in which expression in somatic cells is unusual. However, somatic expression of piRNAs and PIWI proteins has been reported for *Hydra* and *Nematostella*[66,67], and Piwi knockdown in *Hydra* and *Nematostella* affected regeneration and development[67,68], so somatic expression could be a general cnidarian feature. Indeed, pan-arthropod analyses also revealed that somatic piRNAs could be an ancestral defence against viruses[69]. It is unclear whether somatic expression in cnidarians is related to defence, or whether it reflects imperfect segregation of germline and soma where somatic cells can contribute to future generations, implying protection from transposable element activity is necessary.

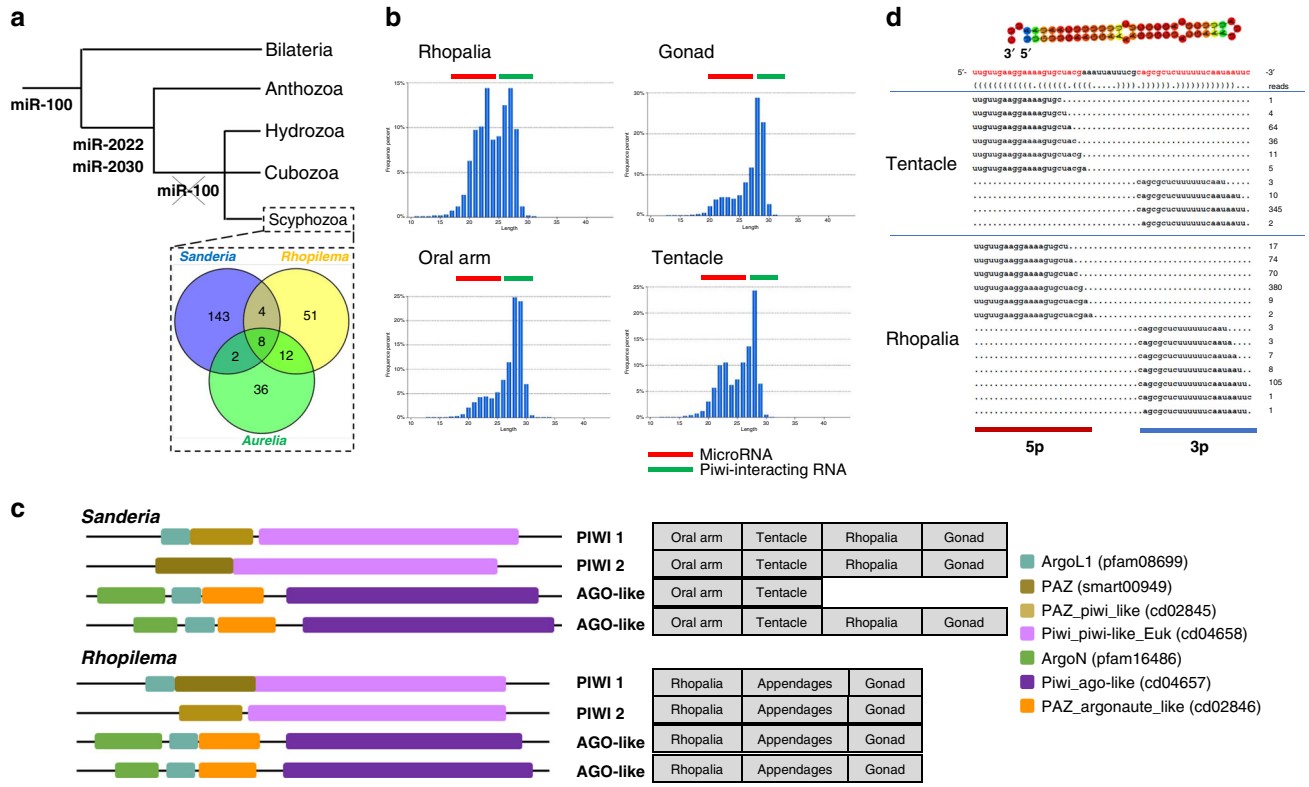

**Fig. 5 Small RNAs in jellyfishes. a** Gain and loss of microRNAs in bilaterians and cnidarians. miR-100 has been lost in the medusozoan lineages; **b** Size distribution of small RNA reads from different tissues of *Sanderia malayensis*; the red and green bars indicate putative microRNAs and piwi-interacting RNAs, respectively. The presence of both types of small RNAs in somatic tissues is a widespread cnidarian feature; **c** Domain organisation of deduced Ago and PIWI proteins in the two jellyfish genomes; sequences given in supplementary information 1; **d** An example of microRNA in *Sanderia malayensis* displaying differential arm usage in different tissues demonstrating microRNA arm switching in cnidarians.

## Discussion

Comparing the two high-quality jellyfish genomes and transcriptomes generated in this study with other cnidarian and bilaterian genomes, several aspects of genomic organisation (homeobox gene), genome composition (mitochondrial integration), gene family origin (hormones), and post-transcriptional regulation (microRNA) are revealed and have changed and/or expanded our views on their biology.

Homeobox genes provide a fruitful line of enquiry for relating genotypic and phenotypic evolution. Comparing the organisation of ANTP class genes between jellyfish and bilaterians reveals that rearrangement and fragmentation of the hypothesised 'mega-homeobox cluster' took very different evolutionary paths in Cnidaria and Bilateria. If distinct Hox and NK gene clusters are associated with ectodermal and mesodermal patterning, respectively, this feature could have evolved in the bilaterian lineage. Further, since a three gene ParaHox gene cluster was ancestral for Bilateria, it is possible that this was present in the Cnidaria-Bilateria common ancestor.

Cnidarians may have linear or circular mtDNA genomes, and this may affect propensity to be copied and integrated into the nuclear genome as NUMTs. Absence of NUMTs is rare in eukaryotes but has been reported for amphioxus and zebrafish, although these species have circular mitochondrial genomes[70]. We show that two jellyfish species have linear mitochondrial genomes but zero NUMTs; this does not support the hypothesis that linear mtDNA is more prone to insertion into the nuclear genomes, or at least is not a conserved feature of cnidarians.

Hormone biosynthesis is important in many aspects of animal physiology, and may have been key to the evolution of metamorphosis in animals. We show that genes for sesquiterpenoid hormone production, typical of arthropods, are also present in cnidarians. This suggests the cnidarian-bilaterian ancestor already had an established sesquiterpenoid system, and opens up the possibility of testing for conserved roles in metamorphosis or developmental transitions in cnidarians and bilaterians.

Small RNAs play roles in post-transcriptional gene regulation and evolution. By examining the small RNAs from three jellyfishes, we give insight into conservation and patterns of gain and loss of microRNAs during cnidarian and bilaterian evolution. Somatic piRNA expression is found in jellyfishes, suggesting a conserved cnidarian feature; while the presence of microRNA arm switching extends the existence of the microRNA arm switching mechanism to the cnidarian-bilaterian ancestor.

The two jellyfish genomes and transcriptomes reconstructed here give insight into genomic characters of the cnidarian-bilaterian ancestor, and the divergent pathways followed by different cnidarian lineages. Furthermore, because cnidarians are an important outgroup to bilaterians, these data increase our understanding of the nature of the long-extinct urbilaterian.

## Methods

**Animal husbandry**. Amakusa jellyfish *S. malayensis* used in this study included wild caught individuals obtained from a local supplier in Hong Kong or provided by the Ocean Park Hong Kong. Edible jellyfish *R. esculentum* used in this study were wild caught, from a local supplier in Hong Kong. Medusae of both species were cultured in circulating artificial seawater (salinity 30 ppt) at room temperature at The Chinese University of Hong Kong. Individuals of *S. malayensis* were not fed for several days after arrival in the laboratory before extracting DNA or RNA. Individuals of *R. esculentum* were fed once per week with newly hatched *Artemia*, and were starved for at least two days before extracting DNA or RNA.

**Genome sequencing**. Genomic DNA was extracted from adult jellyfish and subjected to quality control using gel electrophoresis. Qualifying samples were sent to BGI, Macrogen, and Dovetail Genomics for library preparation and sequencing. In addition, a Chicago library was prepared by Dovetail Genomics using the method described[71]. Briefly, ~500 ng of high molecular weight gDNA (mean fragment length = 55 kb) was reconstituted into artificial chromatin in vitro and fixed with formaldehyde. Fixed chromatin was digested with DpnII, the 5′ overhangs filled in with biotinylated nucleotides, and then free blunt ends were ligated. After ligation, crosslinks were reversed and the DNA purified from protein. Purified DNA was treated to remove biotin that was not internal to ligated fragments. The DNA was then sheared to ~350 bp mean fragment size and sequencing libraries were generated using NEBNext Ultra enzymes and Illumina-compatible adapters. Biotin-containing fragments were isolated using streptavidin beads before PCR enrichment of each library. The libraries were sequenced on an Illumina HiSeq X platform to produce 227 and 153 million 2 × 150 bp paired end reads, which provided 62.19x and 252.77x physical coverage of the genome (1–50 kb pairs), for *S. malayensis* and *R. esculentum*, respectively. Dovetail HiC libraries were also prepared in a similar manner as described previously[72]. Briefly, for each library, chromatin was fixed in place with formaldehyde in the nucleus and then extracted Fixed chromatin was digested with DpnII, the 5′ overhangs filled in with biotinylated nucleotides, and then free blunt ends were ligated. After ligation, crosslinks were reversed and the DNA purified from protein. Purified DNA was treated to remove biotin that was not internal to ligated fragments. The DNA was then sheared to ~350 bp mean fragment size and sequencing libraries were generated using NEBNext Ultra enzymes and Illumina-compatible adapters. Biotin-containing fragments were isolated using streptavidin beads before PCR enrichment of each library. The libraries were sequenced on an Illumina HiSeq X platform to produce 219 and 203 million 2 × 150 bp paired end reads, which provided 1665.28x and 17,177.20x physical coverage of the genome (1–50 kb pairs), for *S. malayensis* and *R. esculentum*, respectively. Details of the sequencing data can be found in Supplementary Tables 1 and 2.

**Transcriptome sequencing**. Transcriptomes of multiple tissues from adult jellyfish of each species were sequenced at BGI. Total RNA from different tissues was isolated using TRIzol reagent (Invitrogen) according to the manufacturer's instructions, and subjected to quality control using a Nanodrop spectrophotometer (Thermo Scientific), gel electrophoresis, and Agilent 2100 Bioanalyzer (Agilent RNA 6000 Nano Kit). Qualifying samples underwent library construction and sequencing at BGI; polyA-selected RNA-Sequencing libraries were prepared using TruSeq RNA Sample Prep Kit v2. Insert sizes and library concentrations of final libraries were determined using an Agilent 2100 bioanalyzer instrument (Agilent DNA 1000 Reagents) and real-time quantitative PCR (TaqMan Probe), respectively. Small RNA (< 200 nt) was isolated using the mirVana miRNA isolation kit (Ambion) according to the manufacturer's instructions. Small RNA was dissolved in the elution buffer provided in the mirVana miRNA isolation kit (Thermo Fisher Scientific) and submitted to BGI for HiSeq Small RNA library construction and 50 bp single-end (SE) sequencing. Details of the sequencing data can be found in Supplementary Tables 4 and 5.

**Sequencing data pre-processing**. For Illumina sequencing data, adapters were trimmed and reads were filtered using the following parameters '-n 0.1 (if N accounted for 10% or more of reads) -l 4 -q 0.5 (if the quality value is lower than 4 and accounts for 50% or more of reads)'. FastQC was run as a quality control[73]. If adapter contamination was identified, adapter sequences were deduced using minion[74]. Adapter trimming and quality trimming was then performed with cutadapt v1.10[75].

**Estimation of genome characteristics**. For each jellyfish species, k-mers of the Illumina PE library of 500 bp insert size were counted using DSK version 2.1.0 with $k = 25$[76], and estimation of genome size, repeat content, and heterozygosity were analysed based on a k-mer-based statistical approach in the GenomeScope webtool[77]. Kraken was used to estimate the percentage of reads that could be contamination from bacteria[78].

**S. malayensis genome assembly**. PacBio long-read data of *S. malayensis* were assembled using FALCON v0.7 and then phased and polished using FALCON_unzip[79]. Pilon was further used to correct indels in the final assembly using the Illumina 500 bp library[80]. The 500 bp library was selected for assembly polishing. The TrimDup module in Rabbit was used to label redundant and heterozygous sequences using default parameters[81]. In addition, Chromium WGS reads were separately used to make a de novo assembly using Supernova (v 2.1.0). Comparison of the two assemblies showed a similar complete BUSCO but higher N50 in the polished PacBio version than in the version made with chromium WGS reads, and hence the PacBio version was used as the input de novo assembly. The de novo assembly, shotgun reads, Chicago library reads, and Dovetail HiC library reads were used as input data for HiRise, a software pipeline designed for using proximity ligation data to scaffold genome assemblies[71]. An iterative analysis was conducted. First, Shotgun and Chicago library sequences were aligned to the draft input assembly using a modified SNAP read mapper (http://snap.cs.berkeley.edu).

The separation of Chicago read pairs mapped within draft scaffolds were analysed by HiRise to produce a likelihood model for genomic distance between read pairs, and the model was used to identify and break putative misjoins, to score prospective joins, and to make joins above a threshold. After aligning and scaffolding Chicago data, Dovetail HiC library sequences were aligned and scaffolded following the same method. After scaffolding, shotgun sequences were used to close gaps between contigs. The mitochondrial genome was assembled using Illumina short reads with SOAPdenovo2[82].

**R. esculentum genome assembly**. For *R. esculentum*, all Illumina short-read sequencing data and PacBio long-read sequencing data were assembled using the hybrid de novo assembly module of MaSuRCA assembler that features a mega-reads algorithm[83]. The resulting assembly was further gap-filled using Gapfiller and PBJelly[84,85]. Owing to observed redundancy of the assembly caused by high heterozygosity, HaploMerger2 was used to construct a representative haploid assembly from the gap-filled assembly[86]. The de novo assembly, shotgun reads, Chicago library reads, and Dovetail HiC library reads were used as input data for HiRise, a software pipeline designed for using proximity ligation data to scaffold genome assemblies[71]. An iterative analysis was conducted. First, Shotgun and Chicago library sequences were aligned to the draft input assembly using a modified SNAP read mapper (http://snap.cs.berkeley.edu). The separation of Chicago read pairs mapped within draft scaffolds were analysed by HiRise to produce a likelihood model for genomic distance between read pairs, and the model was used to identify and break putative misjoins, to score prospective joins, and to make joins above a threshold. After aligning and scaffolding Chicago data, Dovetail HiC library sequences were aligned and scaffolded following the same method. After scaffolding, shotgun sequences were used to close gaps between contigs. The mitochondrial genome was assembled using Illumina short reads with SOAPdenovo2[82]. All analyses were carried out on this version of genome assembly.

**Removal of contaminating sequences**. Assembled contigs were aligned against the mitochondrial genome and assembled bacterial genome sequences to remove contigs that originated from bacteria or mitochondria. To further remove contaminating sequences of unknown origin, we searched against two databases: complete viral genomes (ftp://ftp.ncbi.nih.gov/genomes/Viruses/all.fna.tar.gz) and the NCBI complete and draft bacteria genome assemblies (ftp://ftp.ncbi.nlm.nih.gov/genomes/genbank/bacteria/assembly_summary.txt) databases using blastn[87]. Two cutoff values (alignment length > 500 bp and E-value ≤ 1e-10) were used to identify potential contaminating contigs. For contigs that aligned to NCBI bacteria genome assemblies under these cutoffs, the contig annotation tool CAT was used for their removal from the assembly[88].

**Genome assembly comparison and evaluation**. The completeness of the genome assemblies was assessed using Benchmarking Universal Single-Copy Orthologs (BUSCO) version 3.0, which assess genome completeness using the conserved genes from BUSCO databases[89]. Assembly statistics were visualised using assembly-stats version 1.0.1 (https://github.com/rjchallis/assembly-stats). BUSCO results were compared to that of published cnidarian genomes, using their most recent updated assembly version. The completeness of the jellyfish assemblies reported here was also assessed by mapping of Illumina genomic reads using sequence alignment tool BWA[90]. K-mer duplication levels within an assembly were examined by comparing k-mers found in a randomly selected subset of reads from the Illumina PE library of 500 bp insert size, to k-mers found in that assembly using KAT[91].

**Repeat annotation**. RepeatModeler is a de novo repeat family identification and modelling package containing two de novo repeat-finding programmes RECON and RepeatScout[92,93], and was employed to detect transposable elements (TEs) in the genome using default parameters. For repeat library annotation, three methods were used: (i) the RepeatClassifier module in the RepeatModeler package for classification of identified repetitive elements based on RepBase[94,95]; (ii) tblastx (1e-5) against RepBase version 22.09; (iii) blastx (1e-5) against GyDB v2[96]. The resulting repeat library was then used to estimate repeat compositions in the assemblies using Repeatmasker[97]. LTR retrotransposons in the genomes were further predicted using LTRharvest and LTRdigest[98,99], which detect full length LTR retrotransposons based on their structural features and detect low-copy LTR retrotransposons; these were classified using RepeatClassifier module in the RepeatModeler package. MITEs were predicted in the genomes using MITE-hunter[100]. For repeat-masking prior to gene model prediction, TransposonPSI was also used to identify transposon ORFs, specifically targeting degenerate transposon fragments in the genome[101]. Repeats identified by RepeatModeler and TransposonPSI were used to mask the genome assembly by RepeatMasker prior to gene prediction.

**Gene model prediction**. Raw sequencing reads of the transcriptomes were pre-processed with quality filtering by trimmomatic (v0.33, minimum length 25). For the nuclear genomes, the genome sequences were cleaned and masked by Funannotate, the softmasked assembly were used to run 'funannotate train' with parameters ' --stranded RF–max_intronlen 350,000' to align RNA-seq data, ran

Trinity, and then ran PASA. The PASA gene models were used to train Augustus in 'funannotate predict' step. The gene models were predicted by funannotate predict with parameters '--protein_evidence uniprot_sprot.fasta --genemark_mode ET --busco_seed_species fly --optimize_augustus --busco_db metazoa --organism other --max_intronlen 350000', the gene models from several prediction sources, including 'Augustus', high-quality Augustus predictions (HiQ), 'GeneMark', 'GlimmerHMM', 'pasa', 'snap' were passed to Evidence Modeller (EVM Weights: {'GeneMark': 1, 'HiQ': 2, 'pasa': 6, 'proteins': 1, 'Augustus': 1, 'GlimmerHMM': 1, 'snap': 1, 'transcripts': 1}), and generated the final annotation files, and then used of PASA to update the EVM consensus predictions, added UTR annotations and models for alternatively spliced isoform. The motifs and domains in protein sequences were annotated using InterProScan by searching publicly available databases[102]. For mitochondrial genomes, prokka was used for automatic genome annotation[103], and the results were manually corrected.

**Analysis of different gene families**. Potential gene family sequences were first retrieved from the two genomes using tBLASTn[104]. Identity of each putatively identified gene was then tested by comparison to sequences in the NCBI nr database using BLASTx. For homeobox genes retrieval, sequences were also analysed using the BLAST function in HomeoDB. For phylogenetic analyses of gene families, DNA sequences were translated into amino acid sequences and aligned to other members of the gene family; gapped sites were removed from alignments using MEGA and phylogenetic trees were constructed using MEGA.

**Phylogenomic tree construction and gene family analyses**. Potential orthologues between species were grouped by OrthoMCL[105] with the threshold for BLASTp set as 1e-5. Only genes that were single-copy in each species and identified in at least half of taxa were remained for downstream analyses. This resulted in 268 single-copy orthologues (68,273 residues across 22 species). The species used are shown in the Supplementary Table 8. For each orthologue group, the sequences were aligned with MUSCLE[106], trimmed by trimAl version 1.4[107]. Maximum-likelihood phylogenetic inference was performed by RaxML v8.2.4 with the GTR + Γ model assigned to each partition.

**Synteny analyses**. Synteny blocks between the Hox and ParaHox genes in genomes of *Aurelia baltic*, *Clytia hemisphaerica*, *Morbakka virulenta*, *Nemopilema*, *Nematostella vectensis*, and the two jellyfishes in this study (*Sanderia* and *Rhopilema*) were computed using SyMAP v4.2 (Synteny Mapping and Analysis Programme) with 'mask_all_but_genes = 1' to masked non-genic sequence and other default parameters (Soderlund et al.[108]). Synteny blocks between the Hox and ParaHox genes in genomes of *Homo sapiens*, *Branchiostoma floridae*, and the two jellyfishes *Sanderia* and *Rhopilema* were computed using SyMAP v4.2 with the default parameters except Min Dots = 3 (Minimum number of anchors required to define a syntenic block = 3) and 'mask_all_but_genes = 1' to masked non-genic sequence[108].

**Small-RNA analyses**. Adaptor sequences were trimmed from small RNA-sequencing reads and Phred quality score <20 were removed. Processed reads of length within 18 bp and 27 bp were then mapped to respective genomes using mapper.pl module of the mirDeep2 package[109]. Novel microRNAs were identified using miRDeep2 and were checked manually to fulfil the criteria of MirGeneDB (http://mirgenedb.org/information). The final results were annotated for sequence similarity to known miRNAs in the miRbase[110]. Quantification was produced by the quantifier.pl module of the mirDeep2. Further, to examine the conserveness of miRNA genes in various scyphozoan jellyfish, MapMi[111] was used to identify potential miRNA loci in the genomes of *S. malayensis*, *R. esculentum* and *Aurelia*. The miRNA sequences of *S. malayensis*, *R. esculentum* and *Aurelia* were used as query sequences for MapMi identification of potential miRNA loci in each jellyfish genome, respectively (MapMi scorer cutoff = 15). In addition, the presence of homologous miRNA loci in jellyfish genomes were also examined using miRNA hairpin sequences by carrying Blastn search with *e* < 0.1, followed by checking the hairpin structure with CentroidFold. Multiple alignment of conserved miRNAs was carried out by MEGA7.

**Ethical statement**. We have complied with all relevant ethical regulations for animal research.

**Reporting summary**. Further information on research design is available in the Nature Research Reporting Summary linked to this article.

## Data availability
The final genome assemblies have been deposited on NCBI with accession numbers RQOL00000000 and SWAQ00000000, and the raw genome sequencing reads have been deposited in the NCBI Sequence Read Archive (SRA) under BioProject accession no. PRJNA505074. The mRNA and sRNA transcriptomic data have been deposited on NCBI with accession numbers SRR8193750-8193752, SRR819510-8195102, SRR9590811-95908113, SRR9657804-9657806. All data is available from the corresponding author upon reasonable request.

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

## Acknowledgements

This work was supported by the TUYF Charitable Trust (Grant number: 6903956), Hong Kong Research Grants Council (RGC) General Research Fund (GRF) (14103516, 14100919), and The Chinese University of Hong Kong (CUHK) Direct Grant (BL13937/4053043) to JHLH. We would like to thank Suzanne Gendron and the Aquarium Department of the Ocean Park Hong Kong and other local companies for specimens and photographs; and C.K. Cheung, W.C. Yiu, and A. Leung for help with the extraction of samples.

## Author contributions

W.N. carried out the final genome assembly with the help of T.S., gene model prediction, small RNA mapping, microRNA arm switching detection, small RNA analyses and genome comparison with help of J.H.L.H. J.C. carried out the initial genome assembly, NUMT analyses, and ParaHox gene annotation with help of J.H.L.H. Y.L. carried out the homeobox gene, Ago, and small RNA analyses with help of J.H.L.H. and P.W.H.H. Z.Q. carried out the hormone analyses and small RNA analyses with help of J.H.L.H., W.B. and S.T. J.S. carried out the phylogenomic analyses with the help of P.Y.Q. and J.-W.Q. H.Y.Y. ensured the sequencing project management and animal husbandry. H.S.K., T.F.C., K.Y.Y., K.H.C., S.M.N., K.Y.K.T. contributed the discussion of project at different stages. J.H.L.H. and P.W.H.H. designed and coordinated the project. J.H.L.H., W.N. and P.W.H.H. wrote the initial manuscript; all authors revised and contributed to the final version of the text.

## Competing interests

The authors declare no competing interests.
