## [Peer Review File · Nature Communications]

Reviewers' comments:

Reviewer #1 (Remarks to the Author):

The manuscript by Nong et al describes the sequencing of the genomes of two scyphozoan species, *Sanderia malayensis* and *Rhopilema esculentum*. The sequencing of a genome, while becoming a routine task, is always providing a valuable source for much information. In this case, these are only the 2nd and 3rd scyphozoan jellyfish genomes to become available, after the publication of the moon jellyfish (*Aurelia*) genome published in late 2018. However, as invaluable as these two genome sequences can be, the writing of this manuscript is disappointing, to say the least. The focus, the width and even the arguments presented in this manuscript are limited and confusing. This is not something that can be fixed by changing a word here and there, but requires very significant rewriting and expansion of the manuscript. I still believe that the data generated in this work is invaluable for anyone interested in animal evolution, but it will require a lot of hard work to make this manuscript a worthwhile reading for the scientific community. I am providing remarks in a point-by-point format below and I hope the authors will use them to significantly improve their manuscript:

Major remarks

1. While I can understand the appeal of focusing on the three topics of Hox clusters, mitochondrial genomics and miRNA evolution, I am sure that there are plenty of other interesting topics the authors can cover. A major aspect which is missing in this manuscript is a comparative approach. The authors present here two genomes. Compare them! Are the Hox clusters identical between the scyphozoan species? Are they different? How many miRNA are shared between them, etc. etc. Moreover, the *Aurelia* genome was very recently published (Gold et al. 2019 *Nat Ecol Evol* 3: 96-104), so the authors must compare everything they find at the genomic level of their two species to the *Aurelia* genome. This should be done at detail.
2. Continuing the previous point, the homeobox content of *Aurelia* was already published (see Gold et al. 2019), the authors must compare their results regarding the Homeobox content and structure to the ones from *Aurelia*.
3. Right now the manuscript has only three figures, the first two of them is very simplistic. I strongly believe that it is essential to put the phylogeny (Supplementary information S3) in the main text. Add *Aurelia* to the tree. Explain the construction method of the tree (right now there is no explanation how the tree was constructed).
4. In Fig. 3 the synteny of some of clusters is not so striking. How can we be sure these are all homologous? It's possible that many of duplications are independent. Please provide better and more detailed explanation for the claim these three clusters are homologous to the bilaterian clusters.
5. The authors ignore the fact that the *S. malayensis* genome they determine is only 180 MB, making it one of the smallest cnidarian genomes ever to be reported (*Aurelia*: 700 MB, *Nematostella*: 350 MB, *Aiptasia*: 260 MB, *Acropora digitifera*: 420 MB). What makes this significant size difference? Is it a far smaller number of transposable elements? Shorter introns? Shorter intergenic regions? The authors should at the very minimum identify the sources of this size difference and mention them in the text.
6. The miRNA section is so lacking that it is difficult to assess its quality. The authors must provide in the supplementary data hairpin structures for all the miRNAs they find and the mapping and number of

reads they find for each small RNA that maps to the precursor (basically the output information they get from miRDeep2).

7. The numbers of miRNAs the authors report for the two species is very large compared to the numbers previously reported for other cnidarian species. This raises the suspicion that some of these miRNAs might not be bona fide miRNAs but actually siRNAs that are also found in cnidarians (Krishna et al. 2013 *Nucleic Acids Res* 41: 599-616; Moran et al. 2014 *Genome Res* 24: 651-63). The authors should critically review the output of miRDeep2 and annotate their miRNAs according to the strict rules previously published (see Fromm et al *Annu Rev Genet.* 2015;49:213-42). Any “miRNAs” that do not meet the criteria by Fromm et al should be annotated as siRNAs.

8. The authors must compare the miRNA content of the two jellyfish species and mention how many miRNAs are shared between the two and whether any of the precursors are found in the same genomic loci. This could serve as a strong indication for the authenticity of these miRNAs as functional miRNAs, but not siRNAs, tend to be conserved between species.

9. piRNAs were reported in Cnidaria to not only regulate transposons but also protein-coding genes (Juliano et al 2014 *Proc Natl Acad Sci USA* 111:337-42, Praher et al 2017 *RNA Biol* 14: 1727-1741). Furthermore, Piwi knockdown in Hydra was shown to interfere with regeneration (Juliano 2014) and in *Nematostella* with development (Modepalli et al. 2018 *PLOS Genet* 14(8):e1007590). These important facts and references should be mentioned as somatic expression of piRNAs appears to have in Cnidaria crucial roles far beyond transposon silencing.

10. Continuing the previous point, piRNAs in the soma were shown also in *Nematostella* during larval and adults stages and not only in Hydra (Praher et al 2017 *RNA Biol* 14: 1727-1741). Please correct in the main text and also in the abstract.

Minor points:

11. In the title the authors include “conservation of small RNA processing”, this is a very general and not very clear statement. I guess the authors refer to the differential strand selection, but this cannot be understood from this general statement. Please rephrase.

12. In the abstract, why to convey the misleading message that jellyfish=scyphozoan? This is not very correct as box-jellyfish (cubozoans) are not scyphozoans, and many people refer to hydrozoan medusa also as jellyfish. Please correct.

13. It is not clear in the introduction what is the link the authors propose between cellular properties and arm-switching (differential use of miRNAs). Please clarify this part.

14. In the tree presented in Supplementary Fig. 3.9, the “AGO2” clade is paraphyletic and cannot share one name. This is very misleading. Actually, from this phylogeny it seems that the Argonaute gene duplications that formed “AGO2” happened independently in Scyphozoa and Anthozoa. Please add to this phylogeny the *Aurelia* Argonaute sequences, and break the annotation of “AGO2” into two separate annotations.

15. In *Nematostella* there are no *Anthox8a* and *Antho8b* (*Ax8a* and *Ax8b*) like appears in Fig. 3, but one gene with multiple splice variants (see Moran et al. 2014 *Genome Res* 24: 651-63). Please correct.

16. In the sentence starting with “We identify a ParaHox” (Page 4 paragraph 3) please refer the reader to Fig. 3, as it can be helpful.

17. The circular phylogenetic tree of ANTP class in the supplementary figures is too crowded and sequences names are masking one another. Please fix this problem.

18. The sentence starting with “Assuming the jellyfish miRNAs regulate...” (page 6 middle of 1st paragraph) is not very clear. Please rephrase and elaborate.

19. The authors mention that Lewis et al. (Nat Ecol Evol 2018 2: 174-181) showed that somatic piRNAs in some arthropods regulate transposons. However, this is inaccurate as the paper showed that these somatic transposons actually mostly silence viruses and not transposons. Please correct.

Reviewer #2 (Remarks to the Author):

General comments

This work reports two scyphozoan cnidarian genomes, one at very high quality (Sanderia, scaffold N50 4.68 Mb), the other, Rhopilema at less high quality (N50 212Kb) but still sufficient to be worth reporting. Scyphozoans produce jellyfish, which until recently have not been analyzed at the genomic level, but there have recently been publications/preprints of 3 or so species.

The genomes would represent a valuable resource of a relatively undersampled clade, but I can find no statement of data availability, or the data itself.

The manuscript is light on questions cnidarian/medusozoan biology per se (by which I mean, in a loose sense, 'what is special about jellyfish and how do they work?'). It highlights three results 1) unusual homeobox cluster conservation 2) lack of integration of linear mitochondrial genomes and 3) piRNAs and miRNA arm switching. These are basically interpreted in the light of the cnidarian/bilaterian ancestor, rather than anthozoan/medusozoan.

The analyses do not come across as being generally important. Rather, it seems like a randomly chosen set of examples with no coherent theme. For example, is the fact that 'the underlying mechanism of miRNA arm switching was established before the cnidarian-bilaterian common ancestor' really important? Why? Should it really be one of the foci of an analysis of jellyfish genomes?

The manuscript is also unusually structured in that there are sections of the supplementary information that are hardly referred to in the main text, leaving the reader wondering what to make of them (Actin genes, and Wnt/beta-catenin signalling).

Specific comments

1. My review package contained no link to the genome data or predicted protein sets that I could find, and I can see no reference to data availability of the genome in the manuscript. It's really impossible to review properly without it.

2. I can't see any reference to the gene counts in the main text (apologies if it is there). These data should not be only in the supplementary information.
3. "The 40 ANTP genes are located on just 13 scaffolds (out of 970..." - the significance of this depends entirely on the number of genes per scaffold. Either calculate the significance or delete.
4. "These linkages indicate that the rearrangement and fragmentation of the hypothesized ancestral 'mega-homeobox cluster' took very different evolutionary paths in Cnidaria and Bilateria". Why not independent duplications of the 'UrHox/UrNk' cluster, followed by differential loss? Based on the data presented, it looks far more likely to me than forcing the Cnidarian/Bilaterian ancestor to have a megacluster.
5. 'suggesting distinct selection pressures on gene rearrangement'. It would strengthen the manuscript to present evidence for selection (or lack of selection in one of the lineages), rather than historical accident, which I do not see ruled out.
6. Parahox cluster. The evidence for the medusozoan CDX being orthologous to the bilaterian CDX looks very thin to me, and relies heavily on the block duplication scenario. It would be helpful to more explicitly point out that it looks like a medusozoan innovation rather than accept the status quo. (cf. the phrasing in the abstract).
7. 'distinct genomic cluster for ectodermal and mesodermal patterning are features that were selected for on the bilaterian lineage.'. There is no evidence presented to support selection for these features at all.
8. miRNAs. 'This identified 391 and 132 miRNAs in *S.malayensis* and *R.esculentum*'. These numbers are puzzling. If miRNAs are important, one would not necessarily expect more than double the miRNA content in one species compared to a relatively closely related one. The authors should comment on possible reasons for this, including artefactual detection bias, and state their preferred explanation.
9. '[...] miRNAs that we could confidently assign were jellyfish-specific'. The phrasing around here should be clarified. Jellyfish the life-cycle stage or jellyfish meaning Medusozoa? What is the reference for 'similar to other cnidarians'? How are miRNAs in 'other cnidarians' jellyfish-specific?
10. 'Expression of the PIWI machinery is generally considered important for protection of the germline' - reference needed - 'so expression in somatic cells is unusual' - this doesn't follow at all. Please also distinguish between somatic cells and somatic stem cells.
11. Discussion of gene orthologs. I assume the relevant methods are section 2.12 on P.9 of the supplementary information. 'Identity of each putatively identified gene was then confirmed by reciprocal comparison against the NCBI nr database using BLASTx'. I don't understand exactly what is meant by 'reciprocal comparison' and I am skeptical that such a procedure would work for the

homeobox genes, for instance, of which there is extensive discussion.

12. P.26-45. Supp. Info. Phylogenetic trees of ANTP homeobox genes. These trees all contain one ingroup and one distantly related outgroup (LMX, the same in every tree). Given this structure, all genes in the ingroup will superficially appear to be orthologous. As such, even if the orthologous sequences are correct, the trees themselves are not a valid source of evidence. A specific example: (P.26) the scyphozoan sequences group in a clade of cnidarian 'Barx' sequences, but in trees with all ANTP class genes (my data) this clade does not form a monophyletic group with bilaterian Barx sequences, which are instead sister to the other set of cnidarian sequences in the tree on P.26 (i.e. Ad XP_015764587.1).

13. P.46. Supp. Info. Phylogenetic tree of total ANTP class. This is an exceptionally bad figure. Many of the leaf labels are illegible due to over-printing. Some leaves have no labels attached at all, and some of these would be crucial for assessing whether orthology assignments are correct. The legend refers to purple and black bootstrap values, but I can only see black values.

14. P.8 'as the sponge *A. queenslandica* is the closest species to cnidarians in the trained list' - any bilaterian is closer to Cnidaria than Amphimedon is, in the sense of having shared a common ancestor more recently.

15. P.23 & P.24 - both tables include TALE class homeobox genes as well as ANTP, but are only labelled 'ANTP class'.

16. Supplementary information, all phylogenetic trees. Is there really any point to giving Neighbor-joining bootstrap values?

17. P.74 Legend states "Bootstrap values shown in blue and black" but only black values are shown.

Reviewer #3 (Remarks to the Author):

The authors conducted genome and transcriptome sequencing of two jellyfish" species (Scyphozoans), which resulted in one high quality genome assembly (*Sanderia malayensis*), and one average quality assembly for its size (*Rhopilema esculentum*). The authors also performed detailed analyses on Hox gene clusters and small RNAs giving insights into the animal evolution.

Given that three other jellyfish genomes (one Hydrozoan, two Scyphozoans) have been publicly available, the novelty of the work required for a publication in Nature Communications is questionable. The previously published jellyfish works (Kim et al 2018, Leclere et al 2018, and Gold et al 2018) are cited but not effectively mentioned in the manuscript to support/elaborate their findings.

Most of the comparisons are done against non-jellyfish Cnidarians, but comparisons within jellyfish or

Scyphozoa could also provide important insight into differences and shared traits among species. For example, 3.1 Heterozygosity, 3.2 Repeat content, etc. should include metrics from other Scyphozoans (by the way, it is difficult to refer to the supplementary tables and figures because no table numbers or figures are assigned).

The authors state that the quality of the genome assemblies presented are higher than other Cnidarian genomes published, but it is not clearly demonstrated how this difference contributed to their findings. Did the difference in the assembly quality between *S. malayensis* and *R. esculentum* (or other previously published Cnidarian genomes) result in qualitatively different conclusions that are novel? In my opinion it is essential for the authors to conduct more thorough analyses and comparisons with other previously published Scyphozoan genomes, at least with the one published in a peer reviewed journal (Gold et al 2018), before being considered for publication in Nature Communications.

Response to Reviewer #1

1) The manuscript by Nong et al describes the sequencing of the genomes of two scyphozoan species, *Sanderia malayensis* and *Rhopilema esculentum*. The sequencing of a genome, while becoming a routine task, is always providing a valuable source for much information. In this case, these are only the 2nd and 3rd scyphozoan jellyfish genomes to become available, after the publication of the moon jellyfish (*Aurelia*) genome published in late 2018. However, as invaluable as these two genome sequences can be, the writing of this manuscript is disappointing, to say the least. The focus, the width and even the arguments presented in this manuscript are limited and confusing. This is not something that can be fixed by changing a word here and there, but requires very significant rewriting and expansion of the manuscript. I still believe that the data generated in this work is invaluable for anyone interested in animal evolution, but it will require a lot of hard work to make this manuscript a worthwhile reading for the scientific community. I am providing remarks in a point-by-point format below and I hope the authors will use them to significantly improve their manuscript: Major remarks

1. While I can understand the appeal of focusing on the three topics of Hox clusters, mitochondrial genomics and miRNA evolution, I am sure that there are plenty of other interesting topics the authors can cover. A major aspect which is missing in this manuscript is a comparative approach. The authors present here two genomes. Compare them! Are the Hox clusters identical between the scyphozoan species? Are they different? How many miRNA are shared between them, etc. etc. Moreover, the *Aurelia* genome was very recently published (Gold et al. 2019 Nat Ecol Evol 3: 96-104), so the authors must compare everything they find at the genomic level of their two species to the *Aurelia* genome. This should be done at detail.

Response 1:

We thank the reviewer for the understanding and positive comments, commenting that our generated data are invaluable. To summarise, Reviewer 1 has asked for (a) expansion of the case studies within the paper; (b) additional comparisons between the two species studied; (c) comparison to *Aurelia*. All these points have been carefully considered.

In this revised version of manuscript, we have (1) improved the genome quality to near-chromosomal level for one species, (2) carried out extensive new analyses to examine and compare the additional gene families in the jellyfish genomes (including the jellyfish genomes of *Aurelia*, *Clytia*, *Morbakka*, *Nemopilema* and our two jellyfish genomes), (3) compared the microRNAs between the two jellyfish species studied here plus moon jellyfish, which required us to generate new microRNA data for *Aurelia*. Further, we have also reorganised the manuscript. Details relating to each suggestion/comment can be found below point-by-point.

2) Continuing the previous point, the homeobox content of *Aurelia* was already published (see Gold et al. 2019), the authors must compare their results regarding the Homeobox content and structure to the ones from *Aurelia*.

Response 2:

We have followed the suggestions of the referee, and analyses of all homeobox gene content have been carried out using all jellyfish genomes (Figure 2, and supplementary information S1, section 1.3.5 and 1.3.6). They are now thoroughly compared in numbers as in published study Gold et al 2019 (supplementary information Table 1.3.7).

Further, we have also carried out detailed analyses of the ANTP-class. As shown in Figure 2 and supplementary information Figure S1.3.1 and Table 1.3.7, the homeobox gene numbers are similar between all jellyfish genomes, but the clustered and interdigitated genomic organisation of Hox and NK genes can only be observed in our two jellyfish genome assemblies, due to the higher scaffold size obtained in the present study.

3) Right now the manuscript has only three figures, the first two of them is very simplistic. I strongly believe that it is essential to put the phylogeny (Supplementary information S3) in the main text. Add Aurelia to the tree. Explain the construction method of the tree (right now there is no explanation how the tree was constructed).

Response 3:

Based on the new experiments and analyses carried out in this version of the manuscript (see response to editor), we now include more data analyses and figures in the paper (1 table plus 6 main figures now).

The phylogenomic tree has also been rebuilt as suggested with the new jellyfish genomes, and included in Figure 1D. The methodology of constructing the tree is stated in the Supplementary information S1 (section 1.2.13).

4) In Fig. 3 the synteny of some of clusters is not so striking. How can we be sure these are all homologous? It's possible that many of duplications are independent. Please provide better and more detailed explanation for the claim these three clusters are homologous to the bilaterian clusters.

Response 4:

We understand that assigning orthology between bilaterian and non-bilaterian homeobox genes is sometimes difficult, and is complicated by the limited length of sequence alignment possible.

In our gene tree analyses, we have observed orthologous genes that have high support values (in either the HoxL, NKL, or also in ANTP trees, Supplementary information Figure S1.3.2-1.3.4) between bilaterian and cnidarian orthologues, including Dbx, Dlx, Evx, Hhex, Meox, Msx, NK1, NK3, and NK6.

The referee is correct that there are cases where orthologues do not have strong support values between bilaterians and cnidarians (which is also commonly shown in other studies). Nevertheless, they still have good support values among cnidarians.

Further, an additional source of information can be synteny, and hence to address this concern, we have carried out syntenic analyses (Supplementary information S1 Figure S1.3.7),

including BarxL, EmxL, NK2/4, NotoL, Msx, MsxL, NK3, NK5, NK7, and TlxL loci, and jellyfish Hox and ParaHox loci to bilaterians clusters (specifically human and amphioxus *Brachiostoma floridae*, this is in addition to syntenic analyses between cnidarian genomes already presented (Figure 2C)). Congruent with previous synteny analyses comparing the sea anemone *Nematostella vectensis* to bilaterian genomes (Putnam et al 2007 *Science* 317(5834):86-94; Hui et al 2008 *Evol Dev* 10(6):725-30), our analyses solidly support homology of ‘sets’ of homeobox genes between bilaterians and cnidarians.

5) The authors ignore the fact that the *S. malayensis* genome they determine is only 180 MB, making it one of the smallest cnidarian genomes ever to be reported (Aurelia: 700 MB, Nematostella: 350 MB, Aiptasia: 260 MB, Acropora digitifera: 420 MB). What makes this significant size difference? Is it a far smaller number of transposable elements? Shorter introns? Shorter intergenic regions? The authors should at the very minimum identify the sources of this size difference and mention them in the text.

Response 5:

We thank the reviewer for this suggestion. In an attempt to address this question, we have carried out additional analyses on 1) repeat content including transposable elements (Supplementary information S1.3.2, Table 1.3.1, Table 1.3.2), 2) exon and intron lengths (Supplementary information S1.3.3), and 3) expanded or deleted gene families in any jellyfish lineages (Supplementary information S5). We found that the jellyfish *Sanderia* genome contains a similar number of predicted genes (27,365) to other cnidarian genomes (ranging from 21,862 to 38,007), but the average gene size (including UTRs deduced from transcriptome reads) is the smallest among all cnidarian genomes (i.e. ~4.5kb per gene). In total, the length of DNA sequence contributing to coding genes in the *Sanderia* genome is small (~123Mb) compared to other cnidarian genomes. The only (high quality) cnidarian genome known with less DNA sequence contributing to genes is the sea anemone *Nematostella* genome (~114Mb); however, in the latter case this comprises only ~32% of its genome in comparison to the ~67% in *Sanderia*. In summary, both small total gene size and small intergenic distances are the major factors contributing to the small genome size of jellyfish *Sanderia*. This information is now included in Supplementary information Table 1.3.3.

6) The miRNA section is so lacking that it is difficult to assess its quality. The authors must provide in the supplementary data hairpin structures for all the miRNAs they find and the mapping and number of reads they find for each small RNA that maps to the precursor (basically the output information they get from miRDeep2).

Response 6:

We have now included the hairpin structures for all miRNAs and their reads in the Supplementary information S2, S3, S4, and S5 as requested.

7) The numbers of miRNAs the authors report for the two species is very large compared to the numbers previously reported for other cnidarian species. This raises the suspicion

that some of these miRNAs might not be bona fide miRNAs but actually siRNAs that are also found in cnidarians (Krishna et al. 2013 Nucleic Acids Res 41: 599-616; Moran et al. 2014 Genome Res 24: 651-63). The authors should critically review the output of miRDeep2 and annotate their miRNAs according to the strict rules previously published (see Fromm et al Annu Rev Genet. 2015;49:213-42). Any “miRNAs” that do not meet the criteria by Fromm et al should be annotated as siRNAs.

Response 7:

We have now re-annotated all miRNAs based on the very conservative criteria suggested (Supplementary information S1-S5). Using these highly stringent criteria, which might underestimate numbers, the numbers for *Sanderia* and *Rhopilema* dropped to 125 and 41 respectively. Thus, we have also included the 24 most microRNA-like candidates (not included here as definite microRNAs) for each species in Supplementary information S2, S3, and S5 such that readers interested in these less confident candidates can study them.

8) The authors must compare the miRNA content of the two jellyfish species and mention how many miRNAs are shared between the two and whether any of the precursors are found in the same genomic loci. This could serve as a strong indication for the authenticity of these miRNAs as functional miRNAs, but not siRNAs, tend to be conserved between species.

Response 8:

This is a good suggestion. In order to carry out a more thorough analyses, we have performed additional small RNA transcriptome sequencing on the original species (and also on the moon jellyfish *Aurelia*). They have now been carefully annotated in suggestion/response 7, and thoroughly compared, and we now revealed the gain and loss of microRNAs between cnidarians and bilaterians, as well as among cnidarians (Figure 6A).

Further, we have also aligned their sequences, as well as carried out the synteny as suggested (supplementary information S1.3.10). A total of 26 microRNAs were found to be present in more than one jellyfish genome, including 2 microRNAs conserved among all cnidarians, 6 microRNAs only conserved among all 3 jellyfish species, 4 microRNAs only conserved between jellyfish *Sanderia* and *Rhopilema*, 12 microRNAs only conserved between jellyfish *Rhopilema* and *Aurelia*, and 2 microRNAs only conserved between *Aurelia* and *Sanderia*. These comparisons solidly confirmed their identity and homology.

9) piRNAs were reported in Cnidaria to not only regulate transposons but also protein-coding genes (Juliano et al 2014 Proc Natl Acad Sci USA 111:337-42, Praher et al 2017 RNA Biol 14: 1727-1741). Furthermore, Piwi knockdown in Hydra was shown to interfere with regeneration (Juliano 2014) and in Nematostella with development (Modepalli et al. 2018 PLOS Genet 14(8):e1007590). These important facts and references should be mentioned as somatic expression of piRNAs appears to have in Cnidaria crucial roles far beyond transposon silencing.

Response 9:

We thank the reviewer for pointing these references, and they have now been included in the manuscript, and shown as follows:

“Somatic expression of piRNAs and PIWI proteins has been reported for *Hydra* and *Nematostella* (Juliano et al 2014; Praher et al 2017), and Piwi knockdown in *Hydra* and *Nematostella* affected regeneration and development (Juliano et al 2014; Modepalli et al 2018), so somatic expression could be a general cnidarian feature.”

10) Continuing the previous point, piRNAs in the soma were shown also in Nemaostella during larval and adults stages and not only in Hydra (Praher et al 2017 RNA Biol 14: 1727-1741). Please correct in the main text and also in the abstract.

Response 10:

It has now been amended in the abstract and main text, and shown as follows:

Abstract: “Somatic and germ-line cells express both piwi-interacting RNAs in jellyfish, a conserved cnidarian feature; we also detect evidence for tissue-specific miRNA arm-switching as found in Bilateria.”

Main text: “Somatic expression of piRNAs and PIWI proteins has been reported for *Hydra* and *Nematostella* (Juliano et al 2014; Praher et al 2017), and Piwi knockdown in *Hydra* and *Nematostella* affected regeneration and development (Juliano et al 2014; Modepalli et al 2018), so somatic expression could be a general cnidarian feature.”

11) Minor points: 11. In the title the authors include “conservation of small RNA processing”, this is a very general and not very clear statement. I guess the authors refer to the differential strand selection, but this cannot be understood from this general statement. Please rephrase.

Response 11:

We understand the reviewer’s point. Nevertheless, we are referring to the conservation of small RNA processing not only from microRNA arm switching (i.e. differential strand selection) but also other perspectives, including piRNA expression, conservation and gain/loss of microRNA members, so we would like to keep the title unless the reviewer has strong concerns.

12) In the abstract, why to convey the misleading message that jellyfish=scyphozoan? This is not very correct as box-jellyfish (cubozoans) are not scyphozoans, and many people refer to hydrozoan medusa also as jellyfish. Please correct.

Response 12:

The sentence is now changed to “The Phylum Cnidaria represents a close outgroup to Bilateria and includes familiar animals such as sea anemones, corals (Anthozoa), hydroids (Hydrozoa), box jellyfish (Cubozoa), and true jellyfish (Scyphozoa).”

13) It is not clear in the introduction what is the link the authors propose between cellular properties and arm-switching (differential use of miRNAs). Please clarify this part.

Response 13:

This part has now been clarified as:

“Numerous individual genes can confidently be deduced to have been present in the urbilaterian (Paps and Holland 2018), as can evolution of post-transcriptional regulation in cellular properties such as mechanisms for differential use of arms from a miRNA duplex (Griffith-Jones et al 2011; Hui et al 2013).”

14) In the tree presented in Supplementary Fig. 3.9, the “AGO2” clade is paraphyletic and cannot share one name. This is very misleading. Actually, from this phylogeny it seems that the Argonaute gene duplications that formed “AGO2” happened independently in Scyphozoa and Anthozoa. Please add to this phylogeny the Aurelia Argonaute sequences, and break the annotation of “AGO2” into two separate annotations.

Response 14:

Other jellyfish Ago sequences have now been included in the new gene tree reconstruction (supplementary information S1, section 1.3.9, Figure S1.3.20). Based on our new analyses, we agree with the referee, and renamed the Ago sequences as Ago-like proteins.

15) In Nematostella there are no Anthox8a and Antho8b (Ax8a and Ax8b) like appears in Fig. 3, but one gene with multiple splice variants (see Moran et al. 2014 Genome Res 24: 651-63). Please correct.

Response 15:

Thank you. Figure 3 has now been amended as suggested.

16) In the sentence starting with “We identify a ParaHox” (Page 4 paragraph 3) please refer the reader to Fig. 3, as it can be helpful.

Response 16:

This has now been included, and rewritten as:

“A ParaHox gene cluster containing three homeobox genes was also identified in both jellyfish species (Figure 2B, Supplementary Information S1), rather than a single ParaHox gene or a two-gene cluster previously reported for Cnidaria (Ryan et al 2007; Hui et al 2008; DuBuc et al 2012).”

17) The circular phylogenetic tree of ANTP class in the supplementary figures is too crowded and sequences names are masking one another. Please fix this problem.

Response 17:

We have now also presented trees in another style, labelled with colour, as well as only showing the values on the major nodes, such that reader can easily refer the sequences (Supplementary information S1, section 1.3.5, Figure S1.3.4).

18) The sentence starting with “Assuming the jellyfish miRNAs regulate...” (page 6 middle of 1st paragraph) is not very clear. Please rephrase and elaborate.

Response 18:

We have now deleted this part from the main text.

19) The authors mention that Lewis et al. (Nat Ecol Evol 2018 2: 174-181) showed that somatic piRNAs in some arthropods regulate transposons. However, this is inaccurate as the paper showed that these somatic transposons actually mostly silence viruses and not transposons. Please correct.

Response 19:

Thank you. The sentence is now amended as:

“Indeed, pan-arthropod analyses also revealed that somatic piRNAs could be an ancestral defence against viruses (Lewis et al 2018).”

Response to Reviewer #2

20) General comments - This work reports two scyphozoan cnidarian genomes, one at very high quality (Sanderia, scaffold N50 4.68 Mb), the other, Rhopilema at less high quality (N50 212Kb) but still sufficient to be worth reporting. Scyphozoans produce jellyfish, which until recently have not been analyzed at the genomic level, but there have recently been publications/preprints of 3 or so species. The genomes would represent a valuable resource of a relatively undersampled clade, but I can find no statement of data availability, or the data itself. The manuscript is light on questions cnidarian/medusozoan biology per se (by which I mean, in a loose sense, 'what is special about jellyfish and how do they work?'). It highlights three results 1) unusual homeobox cluster conservation 2) lack of integration of linear mitochondrial genomes and 3) piRNAs and miRNA arm switching. These are basically interpreted in the light of the cnidarian/bilaterian ancestor, rather than anthozoan/medusozoan. The analyses do not come across as being generally important. Rather, it seems like a randomly chosen set of examples with no coherent theme. For example, is the fact that 'the underlying mechanism of miRNA arm switching was established before the cnidarian-bilaterian common ancestor' really important? Why? Should it really be one of the foci of an analysis of jellyfish genomes? The

manuscript is also unusually structured in that there are sections of the supplementary information that are hardly referred to in the main text, leaving the reader wondering what to make of them (Actin genes, and Wnt/beta-catenin signalling).

Response 20:

We thank the reviewer for the understanding and positive comments on the value and quality of our generated data. The reviewer asks primarily for (a) clarity on data availability, (b) more biological information about true jellyfish, and (c) justification of why we chose particular characteristics to study (homeobox genes, mitochondrial genome integrations, microRNAs).

On (a), data availability, we apologize that the open availability and download route was not made sufficiently clear; please see below response 21 for access details.

On (b), background information, we have now written the following paragraphs showing jellyfish biology and significance,

“Jellyfish is the common name for the free-swimming of gelatinous animals with bells and tentacles, especially the medusa phase of cnidarians although the term is sometimes extended to ctenophores. Within cnidarians, the scyphozoans are sometimes referred to as ‘true jellyfish’, and like other cnidarians their body is constructed from two germ layers and their tentacles are armed with nematocysts with venom for capturing prey and/or defending against predators. Scyphozoans play significant ecological roles from surface waters to the deep sea, as an important part of the oceanic food chain, and they are found in every major ocean in the world. Scyphozoan jellyfish in coastal seas interact with humans in several ways. Thousands of swimmers are stung with varying degrees of severity every year. In addition, when their living conditions are favourable, scyphozoans can form swarms (jellyfish blooms), which can damage fishing apparatus or clog the cooling systems of power stations. Some species in the order Rhizostomae have been adopted as a food source in some regions and are farmed in aquaculture systems.”

On (c), choice of topics, it is inevitable that each scientist will choose different topics. We chose homeobox genes because they are well-studied and a fruitful line of enquiry for relating genotypic and phenotypic evolution. We chose mitochondrial integration, because cnidarians have unusual mtDNA genomes whose evolution is poorly understood. We chose microRNAs because they are one component of gene regulation, and the evolution of gene regulation is of broad interest. We have now added two further case studies: hormone biosynthesis as a topic of general physiological interest, and toxin genes which are of practical relevance as they may relate to which species can be eaten by humans. These systematic analyses of all genes/gene families have provided both breadth and depth for this study. In addition, these two high quality cnidarian genomes provide data are made available for other researchers to examine additional topics of interest. Nevertheless, based on the reviewer’s suggestions, we have now also made the significance of our chosen topics clearer and better connected at the beginning of each sub-section.

The reviewer also comments that one of the jellyfish genomes we report (*Sanderia*) is at “very high quality”, and the other (*Rhopilema*) is lower quality but “still worth reporting. We would like to note that we have now improved the *Rhopilema* genome to a close to

chromosomal level standard (N50=12.93Mb), making these two scyphozoan genomes both at very high quality.

21) Specific comments: 1. My review package contained no link to the genome data or predicted protein sets that I could find, and I can see no reference to data availability of the genome in the manuscript. It's really impossible to review properly without it.

Response 21:

We apologise for not making this clear. We have indeed submitted the genomic assemblies to the NCBI with accession numbers stated in Table 1. In addition, to follow the policy of the journal, we have also submitted all transcriptomic resources to public databases as well (information detailed in Supplementary information S1 section 1.1.3-1.1.5).

For reviewing purposes, we have now created a link to all the data set that the reviewer can access at: <https://drive.google.com/open?id=1uJAICBArRyzqAc18WvHIXuaZCp4pMuTA>.

22) I can't see any reference to the gene counts in the main text (apologies if it is there). These data should not be only in the supplementary information.

Response 22:

The reviewer is correct that the information was only in the supplementary information in the previous version of the manuscript. We have now included the gene counts in the main text in addition to the supplementary information (Supplementary information S1, Table 1.3.3).

23) "The 40 ANTP genes are located on just 13 scaffolds (out of 970..." - the significance of this depends entirely on the number of genes per scaffold. Either calculate the significance or delete.

Response 23:

We have now deleted the sentence as suggested.

24) "These linkages indicate that the rearrangement and fragmentation of the hypothesized ancestral 'mega-homeobox cluster' took very different evolutionary paths in Cnidaria and Bilateria". Why not independent duplications of the 'UrHox/UrNk' cluster, followed by differential loss? Based on the data presented, it looks far more likely to me than forcing the Cnidarian/Bilaterian ancestor to have a megacluster.

Response 24:

This is an interesting idea, but in either scenario the implication is that one (or a few) Hox genes were linked to one (or a few) NK genes in the bilaterian-cnidarian ancestor. The question is how many. To better understand the evolutionary history, we have compared the genes neighbouring the ANTP-class homeobox genes in different cnidarian genomes and bilaterian genomes (supplementary information S1 Figure S1.3.7), as well as among different

cnidarian genomes (Figure 2C). Clear syntenic relationships are revealed, suggesting the homeobox loci are orthologous in the bilaterians and cnidarians. The issue then becomes whether the actual homeobox clusters within these loci expanded prior to the cnidarian/bilaterian ancestor, and what their organisation was. Improved trees (see above) confirm orthology of many of the ANTP class genes between cnidarian and bilaterians: in our analyses *Dbx*, *Dlx*, *Evx*, *Hhex*, *Meox*, *Msx*, *NK1*, *NK3*, and *NK6* are most clear. This indicates that at least some of the NK and Hox-like tandem duplications occurred before the split of cnidarians and bilaterians.

In the original version of the manuscript, it was only the jellyfish *Sanderia* genome (not any other cnidarian genome) that had sufficient genome assembly quality to infer interdigitation of Hox and NK genes. We have now improved the jellyfish *Rhopilema* genome markedly and reinvestigated homeobox genomic locations. Interdigitation of Hox and NK genes is seen in *Rhopilema* genome as well as *Sanderia*. Thus, it is most parsimonious to infer that there were extensive NK and Hox-like gene duplications before the last common ancestor of cnidarians and bilaterians, forming a ‘megacluster’, and the two types of genes became interdigitated somewhere along the lineage leading to true jellyfish.

25) 5. 'suggesting distinct selection pressures on gene rearrangement'. It would strengthen the manuscript to present evidence for selection (or lack of selection in one of the lineages), rather than historical accident, which I do not see ruled out.

Response 25:

We have now deleted the sentence, and rewritten as:

Main text: “These linkages indicate that rearrangement and fragmentation of the hypothesized ancestral ‘mega-homeobox cluster’ took very different evolutionary paths in Cnidaria and Bilateria.”

Main text: “Thus, the contrast organisation of ANTP class genes between jellyfish and bilaterians reveals similarities and differences, with the data suggesting that distinct genomic clusters for ectodermal (Hox) and mesodermal (NK) patterning are features that evolved on the bilaterian lineage.”

26) 6. Parahox cluster. The evidence for the medusozoan CDX being orthologous to the bilaterian CDX looks very thin to me, and relies heavily on the block duplication scenario. It would be helpful to more explicitly point out that it looks like a medusozoan innovation rather than accept the status quo. (cf. the phrasing in the abstract).

Response 26:

It is hard to distinguish between the two scenarios (3 ancestrally, or 2 ancestrally), but either way there are three ParaHox genes in jellyfish and in bilaterians. We have now rewritten the two sections as follows, with the main text clearly pointing out two scenarios:

Abstract: “We describe homeobox gene clusters characterised by interdigitation of Hox, NK, and Hox-like genes revealing an alternate pathway of ANTP class gene dispersal; however, one feature comparable to many bilaterians is a three gene ParaHox cluster.”

Main text: “Our analyses suggest the cluster includes likely orthologues of Pdx and Gsx, with the third gene being either Cdx or an independent duplication. Orthology to cnidarian and bilaterian ParaHox gene regions were also confirmed by syntenic analysis using nearby genes (Figure 2C, Supplementary Information S1).”

27) 7. 'distinct genomic cluster for ectodermal and mesodermal patterning are features that were selected for on the bilaterian lineage.'. There is no evidence presented to support selection for these features at all.

Response 27:

We have now deleted this sentence.

28) 8. miRNAs. 'This identified 391 and 132 miRNAs in *S.malayensis* and *R.esculentum*'. These numbers are puzzling. If miRNAs are important, one would not necessarily expect more than double the miRNA content in one species compared to a relatively closely related one. The authors should comment on possible reasons for this, including artefactual detection bias, and state their preferred explanation.

Response 28:

We thank the reviewer for pointing this out. We assign microRNAs with support of small RNA sequencing reads mapped to the genome sequences. In the original manuscript, we have less small RNA sequencing depth for the *R. esculentum*. In order to make a thorough comparison, we have carried out additional small RNA sequencing for *R. esculentum* to ensure more miRNAs that are expressed will be recovered. In addition, we have also carried out small RNA sequencing from the moon jellyfish to make a thorough comparison.

Together with the comments 7 and 8 from reviewer 1, we have now re-annotated all microRNAs with strict criteria and high confidence (Supplementary information S2, S3, S4, S5). We are still seeing more microRNAs identified in *S. malayensis* than in *R. esculentum*, but this is probably a reflection of the developmental stages being analysed for small RNAs in the two species. Both numbers will probably be incomplete, one more so than the other.

29) 9. '[...] miRNAs that we could confidently assign were jellyfish-specific'. The phrasing around here should be clarified. Jellyfish the life-cycle stage or jellyfish meaning Medusozoa? What is the reference for 'similar to other cnidarians'? How are miRNAs in 'other cnidarians' jellyfish-specific?

Response 29:

We have now clarified the sentence as:

“Similar to other cnidarians, the vast majority of miRNAs that we could confidently assign were species-specific with only two miRNAs - miR-2022 and miR-2030 - shared with the

anthozoan *N. vectensis* (Figure 6A, Supplementary information S1-3). However, the miR-100 has been lost in the medusozoan lineages (Figure 6A), suggesting non-conserved microRNAs involved in the gene regulation of jellyfish and facilitating the evolution of morphological complexity.”

In addition, to better reflect the situation, we have now included it in Figure 6A.

30) 10. 'Expression of the PIWI machinery is generally considered important for protection of the germline' - reference needed - 'so expression in somatic cells is unusual' - this doesn't follow at all. Please also distinguish between somatic cells and somatic stem cells.

Response 30:

The point we were making originally is that in animals with a germ-soma distinction, the Piwi machinery might be expected to be expressed in germline only, as a protector of the germline. However, as Reviewer 1 pointed out, the Piwi machinery may have additional roles in cnidarians, beyond protection of the germline. We have now rewritten the sentence as:

“Somatic expression of piRNAs and PIWI proteins has been reported for *Hydra* and *Nematostella* (Juliano et al 2014; Praher et al 2017), and Piwi knockdown in *Hydra* and *Nematostella* affected regeneration and development (Juliano et al 2014; Modepalli et al 2018), so somatic expression could be a general cnidarian feature.”

31) 11. Discussion of gene orthologs. I assume the relevant methods are section 2.12 on P.9 of the supplementary information. 'Identity of each putatively identified gene was then confirmed by reciprocal comparison against the NCBI nr database using BLASTx'. I don't understand exactly what is meant by 'reciprocal comparison' and I am skeptical that such a procedure would work for the homeobox genes, for instance, of which there is extensive discussion.

Response 31:

Reciprocal best-hit BLAST is a commonly used method to verify putative gene identities, where one blast a gene sequence of a species and find certain genes of other species above certain threshold in a database, and then take potential homologues and carry out blast again. Although it must be applied with caution for testing orthology, it is robust at assigning genes to gene families. In our experience it also works well for homeobox genes, but in this paper, we have additionally used phylogenetic trees as well as synteny to assign homeobox and other genes to gene family/class/subclass.

For clarity, we have rewritten the method section as:

“Potential gene family sequences were first retrieved from the two genomes using tBLASTn (Altschul, et al. 1990). Identity of each putatively identified gene was then tested by comparison to sequences in the NCBI nr database using BLASTx. For homeobox genes retrieval, sequences were also analysed using the BLAST function in HomeoDB. For phylogenetic analyses of gene families, DNA sequences were translated into amino acid

sequences and aligned to other members of the gene family; gapped sites were removed from alignments using MEGA and phylogenetic trees were constructed using MEGA.”

32) 12. P.26-45. Supp. Info. Phylogenetic trees of ANTP homeobox genes. These trees all contain one ingroup and one distantly related outgroup (LMX, the same in every tree). Given this structure, all genes in the ingroup will superficially appear to be orthologous. As such, even if the orthologous sequences are correct, the trees themselves are not a valid source of evidence. A specific example: (P.26) the scyphozoan sequences group in a clade of cnidarian 'Barx' sequences, but in trees with all ANTP class genes (my data) this clade does not form a monophyletic group with bilaterian Barx sequences, which are instead sister to the other set of cnidarian sequences in the tree on P.26 (i.e. Ad XP_015764587.1).

Response 32:

Similar to suggestion 17 by reviewer 1, we have now also presented trees in another style, labelled with colour, as well as only showing the values on the major nodes, such that reader can easily refer the sequences (Supplementary information S1, section 1.3.5, Figure S1.3.2, S1.3.3, S1.3.4).

Also, we have now rebuilt the HoxL and NKL trees (Supplementary information S1.3.2-3) based on the comment.

Further, we have also carried out syntenic analyses for some of the homeobox genes to confirm their bilaterian homologues, including BarxL and other homeobox genes (Supplementary information S1.3.6, Figure S1.3.7-9).

33) 13. P.46. Supp. Info. Phylogenetic tree of total ANTP class. This is an exceptionally bad figure. Many of the leaf labels are illegible due to over-printing. Some leaves have no labels attached at all, and some of these would be crucial for assessing whether orthology assignments are correct. The legend refers to purple and black bootstrap values, but I can only see black values.

Response 33:

Similar to suggestions 17 and 32, we have now presented trees in another style, labelled with colour, as well as only showing the values on the major nodes, such that reader can easily refer the sequences (Supplementary information S1, section 1.3.5, Figure S1.3.2, S1.3.3, S1.3.4).

34) 14. P.8 'as the sponge *A. queenslandica* is the closest species to cnidarians in the trained list' - any bilaterian is closer to Cnidaria than Amphimedon is, in the sense of having shared a common ancestor more recently.

Response 34:

We have now removed the sentence.

35) 15. P.23 & P.24 - both tables include TALE class homeobox genes as well as ANTP, but are only labelled 'ANTP class'.

Response 35:

They have now been amended.

36) 16. Supplementary information, all phylogenetic trees. Is there really any point to giving Neighbor-joining bootstrap values?

Response 36:

We think that it should do no harm to provide additional data to readers; NJ can be a surprisingly robust method when dealing with large data sets, so we have included both the NJ and ML values for the readers.

37) 17. P.74 Legend states "Bootstrap values shown in blue and black" but only black values are shown.

Response 37:

We thank the reviewer for the careful reading, and have now amended the colour.

Response to Reviewer #3

38) The authors conducted genome and transcriptome sequencing of two jellyfish" species (Scyphozoans), which resulted in one high quality genome assembly (Sanderia malayensis), and one average quality assembly for its size (Rhopilema esculentum). The authors also performed detailed analyses on Hox gene clusters and small RNAs giving insights into the animal evolution. Given that three other jellyfish genomes (one Hydrozoan, two Scyphozoans) have been publicly available, the novelty of the work required for a publication in Nature Communications is questionable. The previously published jellyfish works (Kim et al 2018, Leclere et al 2018, and Gold et al 2018) are cited but not effectively mentioned in the manuscript to support/elaborate their findings.

Response 38:

We think the availability of a hydrozoan genome is relevant: that is a fundamentally different sort of 'jellyfish' very deeply separated in evolution (just like a dogfish is not a lungfish or a starfish). The other 'true jellyfish' genomes (Scyphozoa) are of rather low assembly contiguity compared to the data presented in the current manuscript, and different biological analyses have been performed on them. Thus, our manuscript is a significant step forward in two ways: genome quality and biological interpretation. Concerning comparative analyses, this is a good point, and we have now reanalysed and compared as best we can considering the difference in assembly quality of different genomes. These include the homeobox genes, hormonal genes, toxin genes, NUMTs, and small RNAs.

39) Most of the comparisons are done against non-jellyfish Cnidarians, but comparisons within jellyfish or Scyphozoa could also provide important insight into differences and shared traits among species. For example, 3.1 Heterozygosity, 3.2 Repeat content, etc. should include metrics from other Scyphozoans (by the way, it is difficult to refer to the supplementary tables and figures because no table numbers or figures are assigned).

Response 39:

Similar to suggestion 38, we have compared all the situations in different jellyfish genomes. In addition, we have also added the page, table, and figure numbers in the Supplementary information.

40) The authors state that the quality of the genome assemblies presented are higher than other Cnidarian genomes published, but it is not clearly demonstrated how this difference contributed to their findings. Did the difference in the assembly quality between *S. malayensis* and *R. esculentum* (or other previously published Cnidarian genomes) result in qualitatively different conclusions that are novel? In my opinion it is essential for the authors to conduct more thorough analyses and comparisons with other previously published Scyphozoan genomes, at least with the one published in a peer reviewed journal (Gold et al 2018), before being considered for publication in Nature Communications.

Response 40:

We have treated this comment very seriously, and have carried out further experiments and analyses to substantially improve our manuscript, including:

- 1) improve the genome quality - the jellyfish *Rhopilema* scaffold is now improved to near chromosomal level (genome size = 256 Mb, N50 = 12.93 Mb), representing the highest quality genome assembly thus far for a cnidarian. Together with the jellyfish *Sanderia* genome (genome size = 184 Mb, N50 = 4.6 Mb), they represent the highest quality jellyfish (and cnidarian) genomes allowing us to make comparisons with other species and, more importantly, detect gene clustering that could not be revealed in other studies;
- 2) comparison of original gene families - all gene families are reanalysed and compared in jellyfish genomes, to shed light on both cnidarian and medusozoan evolution;
- 3) comparison of microRNAs - new small RNA transcriptomes have been sequenced (including moon jellyfish *Aurelia*) and thoroughly compared to understand the evolution of microRNAs in jellyfish;
- 4) new gene family analyses - hormonal gene families (not covered in the previous version of the manuscript) have been investigated, and unexpectedly reveal the existence of the sesquiterpenoid pathway in the cnidarian-bilaterian ancestor;
- 5) new toxin experiments - new proteomics of both jellyfishes (*Sanderia* and *Rhopilema*, the latter known as the edible jellyfish) have been performed to investigate whether differences in toxin genes relate to palatability to humans; and
- 6) manuscript restructuring - the manuscript and supplementary information have been reorganised as suggested.

These comparisons between species have identified different dimension of results:

- 1) situations that can only be found in the two high-quality genomes but not in other jellyfish genomes
 - e.g. homeobox genes genomic arrangement
(i.e. the first time in any cnidarian genomes revealing the interdigitation of Hox and NK genes as well as 3-gene ParaHox cluster located on only two scaffolds in both species);
- 2) situations that can only be confirmed by comparing these genomes
 - e.g. conservation of sesquiterpenoid pathway
(i.e. the first time revealing the sesquiterpenoid pathway in cnidarian genomes);
- 3) situations that can only be found in these unique genomes
 - e.g. toxin peptides in jellyfish
(i.e. the first comparison of toxin peptides in “normal” and “edible” jellyfish);
- 4) situations that are not covered in any of the previous jellyfish genomes
 - e.g. small RNAs analyses
(i.e. the first study of small RNA coming three jellyfish species, revealing the first-time conservation and gain/loss of microRNAs in between cnidarians and also to bilaterians with jellyfish data)
(i.e. the first time of showing somatic piRNA expression in jellyfish)
(i.e. the first time of revealing microRNA arm switching in cnidarians)
 - e.g. mitochondrial and nuclear interactions (i.e. NUMTs)
(i.e. the first time of revealing linear mitochondrial genome does not necessarily prone to nuclear integration)

Based all the new analyses and experiments carried out including providing the first chromosomal level genome of cnidarian, we hope that the manuscript can now be considered positively for publication in the *Nature Communications*.

Reviewers' comments:

Reviewer #1 (Remarks to the Author):

In my opinion, the manuscript by Nong et al. titled “Jellyfish genomes reveal distinct homeobox gene clusters and conservation of small RNA processing” has improved on its experimental side, but the writing is still lacking, sometimes even incoherent and essential data is missing from the text or very hard to find. Unfortunately, I do not think the paper can be considered in its present form and I believe it should be thoroughly edited before it can be further considered. I detail my critique below:

Major remarks:

1. The new chapter about putative toxins is extremely weak. It is not clear in Figure 5 what are P1, P2 and P3 and this is not explained at all in the text. Furthermore, how come *Amphimedon queenslandica* has toxins? This sponge is not venomous. *Nematostella vectensis* is mentioned as the species with most toxins, but maybe this is because its venom was functionally studied? (See for example Columbus-Shenkar et al. 2018 eLife 7: e35014). Doesn't it make sense to refer to studies of cnidarian venom? Moreover, the connection the authors try to make between the lowest number of toxins and edibility of *Rhopilema esculentum* does not make a lot of sense as the venom is completely denatured and becomes harmless after cooking. A good example is *Anemonia sulcata*, a sea anemone that is eaten frequently in Galicia (Spain) despite being a highly venomous species. All in all, this is a very poorly written and analyzed chapter. Please consider either to delete it or to greatly expand and improve it.
2. The findings regarding the small genome size of *Sanderia* and the reasons for that (smaller genes, introns I presume? and small intergenic regions) is fascinating. This is barely mentioned in the text. Please expand on this point.
3. While the authors provide now the miRDeep2 output for miRNAs in the supplementary data, it is not clear at all from the text which miRNAs they consider reliable and which don't. This has to be clarified to the reader. Moreover, it is important to provide a specific list of the eight miRNAs shared between *Aurelia*, *Rhopilema* and *Sanderia*.

Minor remarks:

1. In the abstract please use the full term “microRNA” and not the short one “miRNA”.
2. Why hormones and toxins appear in the same section? It makes sense to separate these two topics (if the authors still insist on talking about putative toxins).
3. The authors write: “Cell-to-cell communication in animals provide systematic control of diverse biological activities, which are heritable and under natural selection. One major type of these crucial signaling molecules are hormones...”. Please provide supporting references.
4. The authors write: “The two major classes of small RNAs are microRNAs (21-23 nt) implicated in post-transcriptional gene regulation and piRNAs (>27 nt) involved in suppression of transposable element activity (Aravin et al 2006).” Why miRNAs and piRNAs are the two major classes? siRNAs are found in a much wider phyletic distribution and are also expressed in cnidarians. Please rephrase to “Two of the major classes”.
5. The authors write: “...piRNAs ensuring that mobile DNA is kept in check in the germline (Moran et al 2017)”. This seems to be the wrong reference as it does not deal with piRNAs.

6. Despite acknowledging the error in the original Argonaute supplementary tree in naming the clade Ago2, and fixing it in their supplementary figure, the authors still call it Ago2 in the main text. Please correct.

7. In page 7 please correct "geness" to "genes".

Reviewer #2 (Remarks to the Author):

General points

--

The general presentation of the manuscript, and in particular the supplementary information is improved. The genome assemblies are now both very good.

In my initial review I said that "The manuscript is light on questions cnidarian/medusozoan biology per se" and that "it seems like a randomly chosen set of examples with no coherent theme". For me, the revised version has not substantially addressed this.

The new manuscript contains an additional section on cnidarian hormones and toxic peptides. The subsection on toxic peptides is uninformative, amounting to little more than counts of putative toxins. The choice of the section on steroids and sesquiterpenoids is not well justified and appears an arbitrary selection from a wide array of possible biosynthetic pathways.

Specific points

--

Modify Fig 2B to distinguish between genes that are adjacent on the same scaffold, and those that have non-homeobox genes between them. e.g. with double and triple parallel marks depending on size.

Parahox genes. The Khalturin et al. paper thoroughly describes a 3 gene Parahox cluster. The present manuscript should discuss that result and cite it in this context. Due reference is particularly important given the prominence in the abstract here.

Cnidarian hormones and toxic peptides. "We searched for genes for putative small peptide toxins in the genomes of sponges and cnidarians". What is the relevance of sponges?

Minor

--

Supplementary info, P.13 "as the sponge *A. queenslandica* is the closest species to cnidarians in the

trained list". Point 34 of the rebuttal letter states that this sentence has been removed, but it hasn't.

Typos etc:

P.5 'A ParaHox gene cluster [...] were also identified' should be 'was'

P.6 'Cell-to-cell communication [...] provide' should be 'provides'

P.7 '[...] the enzymes required [...] in vertebrates is evolved' delete 'is'

P.7 'toxin genes'

P.8 'suggesting the possibly of involvement' should be 'possibility' to be grammatical, but I don't think the point is valid.

Reviewer #4 (Remarks to the Author):

The authors have added a significant amount of new data and analysis to the manuscript, thus have essentially addressed all of my original concerns. However, a few minor editorial issues remain (some parts of the manuscript are not reader-friendly). Once these issues have been addressed I am happy to recommend the manuscript for publication.

1. Some figures are too crowded and the characters in the figures are too small. I understand that this tends to be a common issue with sequence data. Please make sure that the final figures are in a vector format (like figure 4C) not in a raster format so that small texts can be zoomed in to make them legible at least on the PC monitor.

2. I don't know if there is a word count limit but figure legends in the main manuscript are not fully descriptive of what is presented in the figures. For example, "Figure 2 B) Schematic summary of ANTP-class homeobox gene arrangement in the jellyfish *S. malayensis*" but the panel contains other species as well. I also believe each figure needs a figure title.

3. I would suggest adding contig N50 in Table 1. As far as I have examined the contig N50 of the genome assemblies presented in this work are an order of magnitude longer than the previously published jellyfish genomes, and worth mentioning.

Response to Reviewer #1

1) In my opinion, the manuscript by Nong et al. titled “Jellyfish genomes reveal distinct homeobox gene clusters and conservation of small RNA processing” has improved on its experimental side, but the writing is still lacking, sometimes even incoherent and essential data is missing from the text or very hard to find. Unfortunately, I do not think the paper can be considered in its present form and I believe it should be thoroughly edited before it can be further considered. I detail my critique below:

Response 1:

We thank the reviewer for the understanding and positive comments, commenting that we have achieved the journal standard on the experimental side. We have thoroughly edited the manuscript based on the suggestions. Details relating to each suggestion/comment can be found below point-by-point.

2) Major remarks: 1. The new chapter about putative toxins is extremely weak. It is not clear in Figure 5 what are P1, P2 and P3 and this is not explained at all in the text. Furthermore, how come *Amphimedon queenslandica* has toxins? This sponge is not venomous. *Nematostella vectensis* is mentioned as the species with most toxins, but maybe this is because its venom was functionally studied? (See for example Columbus-Shenkar et al. 2018 eLife 7: e35014). Doesn't it makes sense to refer to studies of cnidarian venom? Moreover, the connection the authors try to make between the lowest number of toxins and edibility of *Rhopilema esculentum* does not make a lot of sense as the venom is completely denatured and becomes harmless after cooking. A good example is *Anemonia sulcata*, a sea anemone that is eaten frequently in Galicia (Spain) despite being a highly venomous species. All in all, this is a very poorly written and analyzed chapter. Please consider either to delete it or to greatly expand and improve it.

Response 2:

After serious considerations of the comment, we have decided to delete the toxin section as suggested.

3) 2. The findings regarding the small genome size of *Sanderia* and the reasons for that (smaller genes, introns I presume? and small intergenic reasons) is fascinating. This is barely mentioned in the text. Please expand on this point.

Response 3:

We thank the reviewer and have now expanded their descriptions in the main text, which are shown as follows:

“Despite the genome of *S. malayensis* having the smallest cnidarian genome reported to date (Table 1), it contains a similar number of predicted genes (27,365) to other cnidarian genomes (ranging from 21,862 to 38,007)(Supplementary information S1, Table 1.3.3). Concomitant with this, the average size of *S. malayensis* predicted protein coding genes (including UTRs

deduced from transcriptome reads) is smaller than in other cnidarians analysed (~4.5kb per gene). In addition, the mean size of introns is the smallest of all published cnidarian genomes (381bp, Supplementary information S1, Table 1.3.3). In total, the length of DNA sequence contributing to coding genes in the *S. malayensis* genome is small (~123Mb) compared to other cnidarian genomes (Supplementary information S1, Table 1.3.3). The only (high quality) cnidarian genome known with less DNA sequence contributing to genes is the sea anemone *Nematostella vectensis* genome (~114Mb); however, in the latter this comprises only ~32% of the genome in comparison to the ~50% in *S. malayensis*. Thus, small exons and small introns are the major factors contributing to the small genome size of *S. malayensis*.”

4) 3. While the authors provide now the miRDeep2 output for miRNAs in the supplementary data, it is not clear at all from the text which miRNAs they consider reliable and which don't. This has to be clarified to the reader. Moreover, it is important to provide a specific list of the eight miRNAs shared between Aurelia, Rhopilema and Sanderia.

Response 4:

We did present these data, and apologise for not making this clear enough in the main text. In the earlier version of the manuscript, we included sequences of all annotated miRNA sequences found by miRDeep2 followed by manual checking, and those with high confidence (fulfilling all the criteria set in MirGeneDB). Both categories for each species are included in supplementary information S5; for example, on the first sheet of *Aurelia* microRNA sequences, there are 71 identified miRNA sequences, and 22 of them are of high confidence (Notes at the bottom of the table). In addition, we included the numbers and sequence alignment of those shared by more than one species (in Figure 5A and supplementary information S1, Figure S1.3.10). The reviewer is correct that this information should be made clearer to readers. Thus, we now clarify the findings in the main text, as below:

“We sequenced small RNAs from different tissues of the two jellyfish plus the moon jellyfish *Aurelia aurata* (supplementary information S1: Table 1.1.3-1.1.5), and checked authenticity by mapping to genome sequence and testing for predicted hairpin sequences. 71, 65, and 149 putative microRNAs are annotated in *A. aurata*, *R. esculentum*, and *S. malayensis* respectively; of these, 22, 41, and 125 have high confidence fulfilling all criteria in MirGeneDB (supplementary information S5). As with other cnidarians, the majority of confidently assigned microRNAs were species-specific, with only two - miR-2022 and miR-2030 - shared between jellyfish and the anthozoan *N. vectensis* (Figure 5A, supplementary information S1:Figure S1.3.10; supplementary information S2-4). We also note that miR-100 seems to have been lost in the medusozoan lineages (Figure 5A), and a total of 6 microRNAs are conserved across jellyfish genomes only (supplementary information S1: Table 1.3.14).”

We have also made an additional table in the supplementary information S1, Table 1.3.14 to show the sequences of the conserved microRNAs.

5) Minor remarks: 1. In the abstract please use the full term “microRNA” and not the short one “miRNA”.

Response 5:

This has now been changed as suggested.

6) 2. Why hormones and toxins appear in the same section? It makes sense to separate these two topics (if the authors still insist on talking about putative toxins).

Response 6:

Similar to Response 2, we have now deleted the toxin section.

7) 3. The authors write: “Cell-to-cell communication in animals provide systematic control of diverse biological activities, which are heritable and under natural selection. One major type of these crucial signaling molecules are hormones...”. Please provide supporting references.

Response 7:

The paragraph is now rewritten with citations as follows:

“Many cnidarians, including jellyfish, undergo dramatic metamorphosis including budding during asexual reproduction, a process termed strobilation. To date, little is known about factors that regulate cnidarian life cycles (Helm 2018). In a recent study, cnidarian homologues were found for two genes encoding neuropeptide hormones implicated in insect life cycles, and formerly thought specific to insects: eclosion hormone involved in ecdysone-regulated timing of moulting, and bursicon involved in wing expansion during adult emergence (de Oliveira et al 2019). This raises the possibility that other hormonal systems controlling insect metamorphosis could also be conserved in cnidarians.

The hormonal control of insect metamorphosis involves changing interaction between two principal hormonal systems: ecdysone for control of cuticle moulting and sesquiterpenoids (juvenile) hormones implicated in post-embryonic growth and differentiation (Truman 2019). We have focused on the sesquiterpenoid pathway, since in insects sesquiterpenoids control the transition between developmental stages and are essential for reproduction (Cheong et al 2015; Truman and Riddiford 2019; Truman 2019). Derived from an acetate precursor through the mevalonate pathway, farnesyl units are converted to cholesterol and steroid hormones in vertebrates, or into sesquiterpenoid hormones such as juvenile hormone in insects (Schenk et al 2016; Qu et al 2015; 2017; Figure 4A). The biosynthetic pathway is uncharacterized in non-bilaterians.

In the cnidarian genomes investigated here, genes in the sesquiterpenoid biosynthetic pathway identified include Protein farnesyl transferase (FNT), Ste 24 endopeptidase (ZMPSTE24), prenyl protein peptidase (RCE1), isoprenylcysteine carboxymethyl transferase (ICMT), prenylcysteine oxidase (PCYOXIL) and aldehyde dehydrogenase (ALDH)(Figure 4B and C). These enzymes could control the production of the sesquiterpenoid farnesoic acid (FA). FA is a biologically active stimulant of arthropod vitellogenesis (Mak et al 2005), and it has been thought FA is restricted to the protostome lineage (Figure 4B, C, supplementary information S1: Figure S1.3.12). The role of cnidarian FA in either reproduction or

morphogenesis has yet to be determined. Our findings show that genes for sesquiterpenoid hormone production, typical for arthropods, are also found in cnidarians.”

References:

Cheong SP, Huang J, Bendena WG, Tobe SS, Hui JH (2015). Evolution of ecdysis and metamorphosis in arthropods: the rise of regulation of juvenile hormone. *Integrative and Comparative Biology*, 55(5), 878-890.

Cox RM, McGlothlin JW, Bonier F (2016). Evolutionary endocrinology: hormones as mediators of evolutionary phenomena. *Integrative and Comparative Biology*, 56(2), 121-125.

de Oliveira A, Calcino A, Wanninger A (2019). Ancient origins of arthropod moulting pathway components. *eLife*, 8, e46113.

Helm RR (2018). Evolution and development of scyphozoan jellyfish. *Biological reviews of the Cambridge Philosophical Society*, 93(2),1228-1250.

Mak AS, Choi CL, Tiu SH, Hui JH, He JG, Tobe SS, Chan SM. (2005). Vitellogenesis in the red crab *Charybdis feriatius*: Hepatopancreas-specific expression and farnesoic acid stimulation of vitellogenin gene expression. *Molecular Reproduction and Development*, 70(3), 288-300.

McGlothlin JW, Ketterson ED (2008). Hormone-mediated suites as adaptations and evolutionary constraints. *Philosophical Transactions of the Royal Society B: Biological Sciences*, 363(1487), 1611-1620.

Truman JW (2019). The evolution of insect metamorphosis. *Current Biology*, 29, R1252-1268.

Truman JW, Riddiford LM (2019). The evolution of insect metamorphosis: a developmental and endocrine view. *Philosophical Transactions of the Royal Society B*, 374, 20190070.

Zera AJ, Harshman LG, Williams TD (2007). Evolutionary endocrinology: the developing synthesis between endocrinology and evolutionary genetics. *Annual Review of Ecology, Evolution, and Systematics*, 38, 793-817.

8) 4. The authors write: “The two major classes of small RNAs are microRNAs (21-23 nt) implicated in post-transcriptional gene regulation and piRNAs (>27 nt) involved in suppression of transposable element activity (Aravin et al 2006).” Why miRNAs and piRNAs are the two major classes? siRNAs are found in a much wider phyletic distribution and are also expressed in cnidarians. Please rephrase to “Two of the major classes”.

Response 8:

The sentence is now rewritten as suggested and shown as follows:

“Two of the major classes of small RNAs are microRNAs (21-23 nt) implicated in post-transcriptional gene regulation and piRNAs (>27 nt) primarily involved in suppression of transposable element activity (Aravin et al 2006).”

9) 5. The authors write: "...piRNAs ensuring that mobile DNA is kept in check in the germline (Moran et al 2017)". This seems to be the wrong reference as it does not deal with piRNAs.

Response 9:

We thank the reviewer for pointing this out, and have now replaced it with another reference. The rewritten paragraph is shown as follows:

"Both are thought to play major roles in animal evolution, with microRNAs canalizing development through suppression of transcriptional noise thereby facilitating the strength of natural selection (Peterson et al 2009; Christodoulou et al 2010; Berezikov 2011; Moran et al 2017) and piRNAs ensuring that mobile DNA is kept in check in the germline (Barckmann et al 2018)."

Reference:

Barckmann B, El-Barouk M, Pelisson A, Mugat B, Li B, Franckhauser C, Fiston-Lavier AS, Mirouze M, Fablet M, Chambeyron S. (2018). The somatic piRNA pathway controls germline transposition over generations. *Nucleic Acids Research*, 46(18), 9524-9536.

10) 6. Despite acknowledging the error in the original Argonaute supplementary tree in naming the clade Ago2, and fixing it in their supplementary figure, the authors still call it Ago2 in the main text. Please correct.

Response 10:

Thank you for the careful reading. The sentence in the main text has now been amended.

11) 7. In page 7 please correct "geness" to "genes".

Response 11:

This section has been deleted.

Response to Reviewer #2

12) General points -- The general presentation of the manuscript, and in particular the supplementary information is improved. The genome assemblies are now both very good.

Response 12:

We thank the reviewer for the positive comments.

13) In my initial review I said that "The manuscript is light on questions cnidarian/medusozoan biology per se" and that "it seems like a randomly chosen set of examples with no coherent theme". For me, the revised version has not substantially addressed this.

Response 13:

We have clear scientific reasons for choosing the gene families or systems to analyse, and in rewriting some sections we have aimed at providing our rationale for the specific topics so that they don't appear to be 'randomly chosen'.

We chose homeobox genes because they are well-studied and a fruitful line of enquiry for relating genotypic and phenotypic evolution. We chose hormone biosynthesis as a topic of physiological interest, and key to the evolution of metamorphosis in animals. Furthermore, the recent discovery of ecdysone pathway genes in cnidarian (de Oliveira et al 2019), evolutionarily very distant from insects, raises the question of whether the interacting sesquiterpenoid pathway is also widely conserved. We chose mitochondrial integration, because cnidarians have unusual mtDNA genomes whose evolution is poorly understood. We chose microRNAs because they are a component of gene regulation, and the evolution of gene regulation is of broad interest. We have tried to make the significance of our chosen topics clear and connected at the beginning of each sub-section.

To elaborate, we utilised these two genomes in comparison to other published ones, in order to shed light on different aspects from genomic organisation (homeobox gene), genome composition (mitochondrial integration), gene family origin (hormones), and post-transcriptional regulation (microRNA). More importantly, some of our findings are unexpected and have changed and/or expanded our views on their biology.

We approve of diversity in academic writing and agree with the reviewer that there could be other topics of interests. We trust that these two high-quality cnidarian genomes will provide data for other researchers to examine their special lines of enquiry.

Last but not least, we have done our best to answer the specific comments, as well as incorporate the specific suggestions (please see Response 14-23).

14) The new manuscript contains an additional section on cnidarian hormones and toxic peptides. The subsection on toxic peptides is uninformative, amounting to little more than counts of putative toxins. The choice of the section on steroids and sesquiterpenoids is not well justified and appears an arbitrary selection from a wide array of possible biosynthetic pathways.

Response 14:

We thank the reviewer for pointing out the concerns on the toxin section. Similar to Response 2, we have now removed this section.

For the choice of steroids and sesquiterpenoids, similar to Response 7, we have rewritten the beginning of sub-section as:

“Many cnidarians, including jellyfish, undergo dramatic metamorphosis including budding during asexual reproduction, a process termed strobilation. To date, little is known about factors that regulate cnidarian life cycles (Helm 2018). In a recent study, cnidarian homologues were found for two genes encoding neuropeptide hormones implicated in insect life cycles, and formerly thought specific to insects: eclosion hormone involved in ecdysone-regulated timing of moulting, and bursicon involved in wing expansion during adult emergence (de Oliveira et

al 2019). This raises the possibility that other hormonal systems controlling insect metamorphosis could also be conserved in cnidarians.

The hormonal control of insect metamorphosis involves changing interaction between two principal hormonal systems: ecdysone for control of cuticle moulting and sesquiterpenoids (juvenile) hormones implicated in post-embryonic growth and differentiation (Truman 2019). We have focused on the sesquiterpenoid pathway, since in insects sesquiterpenoids control the transition between developmental stages and are essential for reproduction (Cheong et al 2015; Truman and Riddiford 2019; Truman 2019). Derived from an acetate precursor through the mevalonate pathway, farnesyl units are converted to cholesterol and steroid hormones in vertebrates, or into sesquiterpenoid hormones such as juvenile hormone in insects (Schenk et al 2016; Qu et al 2015; 2017; Figure 4A). The biosynthetic pathway is uncharacterized in non-bilaterians.

In the cnidarian genomes investigated here, genes in the sesquiterpenoid biosynthetic pathway identified include Protein farnesyl transferase (FNT), Ste 24 endopeptidase (ZMPSTE24), prenyl protein peptidase (RCE1), isoprenylcysteine carboxymethyl transferase (ICMT), prenylcysteine oxidase (PCYOXIL) and aldehyde dehydrogenase (ALDH)(Figure 4B and C). These enzymes could control the production of the sesquiterpenoid farnesoic acid (FA). FA is a biologically active stimulant of arthropod vitellogenesis (Mak et al 2005), and it has been thought FA is restricted to the protostome lineage (Figure 4B, C, supplementary information S1: Figure S1.3.12). The role of cnidarian FA in either reproduction or morphogenesis has yet to be determined. Our findings show that genes for sesquiterpenoid hormone production, typical for arthropods, are also found in cnidarians.”

References:

- Cheong SP, Huang J, Bendena WG, Tobe SS, Hui JH (2015). Evolution of ecdysis and metamorphosis in arthropods: the rise of regulation of juvenile hormone. *Integrative and Comparative Biology*, 55(5), 878-890.
- Cox RM, McGlothlin JW, Bonier F (2016). Evolutionary endocrinology: hormones as mediators of evolutionary phenomena. *Integrative and Comparative Biology*, 56(2), 121-125.
- de Oliveira A, Calcino A, Wanninger A (2019). Ancient origins of arthropod moulting pathway components. *eLife*, 8, e46113.
- Helm RR (2018). Evolution and development of scyphozoan jellyfish. *Biological reviews of the Cambridge Philosophical Society*, 93(2),1228-1250.
- Mak AS, Choi CL, Tiu SH, Hui JH, He JG, Tobe SS, Chan SM. (2005). Vitellogenesis in the red crab *Charybdis feriatu*s: Hepatopancreas-specific expression and farnesoic acid stimulation of vitellogenin gene expression. *Molecular Reproduction and Development*, 70(3), 288-300.
- McGlothlin JW, Ketterson ED (2008). Hormone-mediated suites as adaptations and evolutionary constraints. *Philosophical Transactions of the Royal Society B: Biological Sciences*, 363(1487), 1611-1620.
- Truman JW (2019). The evolution of insect metamorphosis. *Current Biology*, 29, R1252-1268.
- Truman JW, Riddiford LM (2019). The evolution of insect metamorphosis: a developmental and endocrine view. *Philosophical Transactions of the Royal Society B*, 374, 20190070.

Zera AJ, Harshman LG, Williams TD (2007). Evolutionary endocrinology: the developing synthesis between endocrinology and evolutionary genetics. *Annual Review of Ecology, Evolution, and Systematics*, 38, 793-817.

15) Specific points -- Modify Fig 2B to distinguish between genes that are adjacent on the same scaffold, and those that have non-homeobox genes between them. e.g. with double and triple parallel marks depending on size.

Response 15:

This is a very helpful suggestion. We have now modified Figure 2B as suggested.

16) Parahox genes. The Khalturin et al. paper thoroughly describes a 3 gene Parahox cluster. The present manuscript should discuss that result and cite it in this context. Due reference is particularly important given the prominence in the abstract here.

Response 16:

We thank the reviewer for bringing this up. We have now discussed the result as well as cited it in the main text, and shown as follows:

“A ParaHox gene cluster containing three homeobox genes was also identified in both jellyfish species (Figure 2B, supplementary information S1: Figure S1.3.6), as also described in moon jellyfish *Aurelia* (Khalturin et al. 2019), and distinct from the single ParaHox gene or a two-gene cluster reported for other Cnidaria (Ryan et al 2007; Hui et al 2008; DuBuc et al 2012). Our analyses suggest the cluster includes likely orthologues of Pdx and Gsx, with the third gene being either Cdx or an independent duplication. Orthology to cnidarian and bilaterian ParaHox gene regions were also confirmed by syntenic analysis using nearby genes (Figure 2C, supplementary information S1: Figure S1.3.7). Analyses of available transcriptome and genome data from other Cnidaria indicate that the third ParaHox gene is widespread amongst Medusozoa (including Staurozoa, Cubozoa, Scyphozoa and some Hydrozoa; Figure 2B, supplementary information S1: Table S1.3.13). Clustering of ParaHox genes are only revealed in scyphozoans, including the first intact ParaHox cluster in the *S. malayensis* (Khalturin et al. 2019; Figure 2B). Since a three gene ParaHox gene cluster was ancestral for Bilateria, it is possible that this was present in the Cnidaria-Bilateria common ancestor.”

17) Cnidarian hormones and toxic peptides. "We searched for genes for putative small peptide toxins in the genomes of sponges and cnidarians". What is the relevance of sponges?

Response 17:

The toxin peptides section has now been removed from the manuscript.

18) Minor -- Supplementary info, P.13 "as the sponge A. queenslandica is the closest species to cnidarians in the trained list". Point 34 of the rebuttal letter states that this sentence has been removed, but it hasn't.

Response 18:

Thank you for the careful reading. This is now removed.

19) Typos etc: --- P.5 'A ParaHox gene cluster [...] were also identified' should be 'was'

Response 19:

This is now amended.

20) P.6 'Cell-to-cell communication [...] provide' should be 'provides'

Response 20:

This has now been amended.

21) P.7 '[...] the enzymes required [...] in vertebrates is evolved' delete 'is'

Response 21:

The suggestion has been incorporated.

22) P.7 'toxin genes'

Response 22:

The section of toxin genes has been removed.

23) P.8 'suggesting the possibly of involvement' should be 'possibility' to be grammatical, but I don't think the point is valid.

Response 23:

The sentence has been deleted as suggested.

Response to Reviewer #4

24) The authors have added a significant amount of new data and analysis to the manuscript, thus have essentially addressed all of my original concerns. However, a few minor editorial issues remain (some parts of the manuscript are not reader-friendly).

Once these issues have been addressed I am happy to recommend the manuscript for publication.

Response 24:

We thank the reviewer for the positive comments, and have thoroughly edited the manuscript based on the suggestions. Details relating to each suggestion/comment can be found below point-by-point.

25) 1. Some figures are too crowded and the characters in the figures are too small. I understand that this tends to be a common issue with sequence data. Please make sure that the final figures are in a vector format (like figure 4C) not in a raster format so that small texts can be zoomed in to make them legible at least on the PC monitor.

Response 25:

We thank the reviewer for the suggestion. We have now replaced some figures to ensure all of them can be read. These include Supplementary information S1 Figure S1.3.1, S1.3.3, S1.3.4, S1.3.8.

26) 2. I don't know if there is a word count limit but figure legends in the main manuscript are not fully descriptive of what is presented in the figures. For example, "Figure 2 B) Schematic summary of ANTP-class homeobox gene arrangement in the jellyfish *S. malayensis*" but the panel contains other species as well. I also believe each figure needs a figure title.

Response 26:

Figure titles for each figure have now been added, and the figure legends have also been expanded with necessary details.

27) 3. I would suggest adding contig N50 in Table 1. As far as I have examined the contig N50 of the genome assemblies presented in this work are an order of magnitude longer than the previously published jellyfish genomes, and worth mentioning.

Response 27:

This is a good suggestion, and we have now included the contig N50 in Table 1. And as the reviewer has speculated, both of our jellyfish genomes also have longer contig N50 in an order of magnitude than previously published jellyfish genomes, and indeed, also for all published cnidarians to date.

REVIEWERS' COMMENTS:

Reviewer #1 (Remarks to the Author):

The revised manuscript by Nong et al. is greatly improved compared to the previous versions that I reviewed and I appreciate the efforts the authors invested in this revision. I believe that this manuscript includes interesting and important findings regarding the molecular evolution of jellyfish genome and these are now much more accessible to the reader. Still, I have a few last minor comments that I believe can help the authors make the manuscript ready for publication:

1. Page 3 line 6: please change “cubozoans” to “cubozoan” as only a single species was sequenced.
2. Page 4 lines 23-24: the word “genome” appears twice. Please remove the first one, so the sentence will be “Despite *S. malayensis* having the smallest cnidarian genome reported to 23 date (Table 1),”
3. Page 9 lines 2-4: The species name is “*Aurelia aurita*” and not “*Aurelia aurata*”. Please correct throughout the text.
4. Page 9 line 38- Page 10 line 2: Use either the UK spelling or US spelling but not a mixture of both. It doesn't make sense to write both “defense” and “defence” in two adjacent sentences. Please correct.
5. My last remark is not so minor: I think it's quite problematic to almost not include any information in the materials and methods regarding how technically the work was done. Please try to move at least some of the information from the supplementary data to the materials and methods section in the main text. This would also be more fair towards fellow scientists who invested a lot of work in developing the tools the authors employed in their study (just for example, the authors apparently used miRDeep2 but do not cite Friedlander et al 2011 *Nucleic Acids Research* 40: 37-52, the paper which describes this publicly available tool developed by the Rajewsky group for identifying miRNAs).

Reviewer #2 (Remarks to the Author):

P.4, line 12 - "94.1% complete BUSCO core eukaryotic genes" - is the statement that eukaryotic genes were used for assessment correct? I think it's more normal to report against Metazoa. 94.1% does not occur in the supplementary section covering BUSCO (1.3.3). BUSCO numbers should be reported with the test that was used (dataset, + genome/transcriptome/proteome) and the numbers in table 1 should be comparable.

P7, line 9,10 - '[...] undergo dramatic metamorphosis including budding during asexual reproduction, a process termed strobilation'. Confusing. I understand strobilation (scyphozoans) to be distinct from budding (hydrozoans) in medusa formation at least, and does either qualify as a metamorphosis of the kind observed in insects? This could be clarified, or the paragraph start from the 2nd sentence.

In general the sesquiterpenoid pathway section has been better integrated by raising the possibility of a conserved role in metamorphosis or developmental transitions, but it is noticeable that the authors do not discuss the presence of these components in non-arthropod bilaterians.

Figure 2b: I find the indications of spacing informative. If the authors could clarify whether the triangle is only used when the length is < 100Kbp. At the moment, the interpretation based on the legend is that all of the > 1Mb breaks have no intervening genes in them. Also, on the figure itself '100Kpb' and '1Mpb' - should these be Kbp and Mbp ?

No title is given for reference 74 of the supplementary information.

Separate file: Response to Reviewer Comments

REVIEWERS' COMMENTS:

Reviewer #1 (Remarks to the Author):

1) The revised manuscript by Nong et al. is greatly improved compared to the previous versions that I reviewed and I appreciate the efforts the authors invested in this revision. I believe that this manuscript includes interesting and important findings regarding the molecular evolution of jellyfish genome and these are now much more accessible to the reader. Still, I have a few last minor comments that I believe can help the authors make the manuscript ready for publication:
1. Page 3 line 6: please change “cubozoans” to “cubozoan” as only a single species was sequenced.

Response 1:

This has now been amended.

2) Page 4 lines 23-24: the word “genome” appears twice. Please remove the first one, so the sentence will be “Despite *S. malayensis* having the smallest cnidarian genome reported to 23 date (Table 1),”

Response 2:

The sentence has now been changed as suggested.

3) Page 9 lines 2-4: The species name is “*Aurelia aurita*” and not “*Aurelia aurata*”. Please correct throughout the text.

Response 3:

It has now been amended.

4) Page 9 line 38- Page 10 line 2: Use either the UK spelling or US spelling but not a mixture of both. It doesn't make sense to write both “defense” and “defence” in two adjacent sentences. Please correct.

Response 4:

It has now been amended to “defence”.

5) My last remark is not so minor: I think it's quite problematic to almost not include any information in the materials and methods regarding how technically the work was done. Please try to move at least some of the information from the supplementary data to the materials and methods section in the main text. This would also be more fair towards fellow scientists who invested a lot of work in developing the tools the

authors employed in their study (just for example, the authors apparently used miRDeep2 but do not cite Friedlander et al 2011 *Nucleic Acids Research* 40: 37-52, the paper which describes this publicly available tool developed by the Rajewsky group for identifying miRNAs).

Response 5:

We have now moved the whole materials and methods section to the main text of the manuscript.

Reviewer #2 (Remarks to the Author):

6) P.4, line 12 - "94.1% complete BUSCO core eukaryotic genes" - is the statement that eukaryotic genes were used for assessment correct? I think it's more normal to report against Metazoa. 94.1% does not occur in the supplementary section covering BUSCO (1.3.3). BUSCO numbers should be reported with the test that was used (dataset, + genome/transcriptome/proteome) and the numbers in table 1 should be comparable.

Response 6:

The complete BUSCO in Table 1 have been updated as suggested to report against Metazoa. In addition, the supplementary information, and the result section in the main text has also been updated as follows:

"The *S. malayensis* genome assembly is 184 Mb with a scaffold N50 of 4.6 Mb spanning 970 scaffolds with 26,914 predicted protein coding genes (Table 1). The high physical contiguity is matched by high completeness, with 90.6% complete BUSCO genes (metazoa_odb9 dataset run in genome mode) (Table 1). The *R. esculentum* genome is 256 Mb with a scaffold N50 of 12.9 Mb and 87.1% BUSCO completeness (metazoa_odb9 dataset run in genome mode) with 18,923 predicted protein coding genes (Table 1)."

7) P7, line 9,10 - '[...] undergo dramatic metamorphosis including budding during asexual reproduction, a process termed strobilation'. Confusing. I understand strobilation (scyphozoans) to be distinct from budding (hydrozoans) in medusa formation at least, and does either qualify as a metamorphosis of the kind observed in insects? This could be clarified, or the paragraph start from the 2nd sentence.

Response 7:

The sentences are now rewritten as:

"Many cnidarians undergo dramatic metamorphosis or developmental transitions, including budding in hydrozoans and strobilation in jellyfish. To date, very little is known about factors that regulate cnidarian life cycles (46)."

8) In general the sesquiterpenoid pathway section has been better integrated by raising the possibility of a conserved role in metamorphosis or developmental transitions, but it is noticeable that the authors do not discuss the presence of these components in non-arthropod bilaterians.

Response 8:

This is a good point. We have now included the study in the lophotrochozoan sesquiterpenoid in the results, and explicitly pointing out to readers the possibility of a conserved role in metamorphosis or developmental transitions. The discussion is shown as follows:

“These enzymes could control the production of the sesquiterpenoid farnesoic acid (FA). FA is a biologically active stimulant of arthropod vitellogenesis (53), and it has been thought FA is restricted to the protostome lineage (Figure 4B, C, Supplementary Figure 13). The role of cnidarian FA in either reproduction or morphogenesis has yet to be determined. Previously, sesquiterpenoid methyl farnesoate has been found a non-arthropod bilaterian (annelid *Platynereis dumerilii*)(54). Our findings show that genes for sesquiterpenoid hormone production, typical for arthropods, are also present in cnidarians.”

9) Figure 2b: I find the indications of spacing informative. If the authors could clarify whether the triangle is only used when the length is < 100Kbp. At the moment, the interpretation based on the legend is that all of the > 1Mb breaks have no intervening genes in them. Also, on the figure itself '100Kpb' and '1Mpb' - should these be Kbp and Mbp ?

Response 9:

The figure legend is now amended to include all related information, and shown as follows:

“Figure 2. Homeobox genomic organisation. A) Schematic diagram showing origin of bilaterian homeobox gene clusters from a hypothesized ANTP class megacluster, with ParaHox cluster, Hox cluster, NK cluster, and NK2 genes located on separate chromosomes; B) Schematic summary of ANTP-class homeobox gene arrangement in the jellyfish genomes; ‘?’ denotes divergent ANTP-class homeobox genes; “//” denotes genomic distance more than 100kb and less than 1Mb; “///” denotes genomic distance over 1Mb; triangle denotes intervening non-homeobox genes and is only used when the distance less than 100kb. A three gene ParaHox cluster is present in *S. malayensis*, and the interdigitation of Hox and NK cluster genes is recovered in *Rhopilema esculentum* and *S. malayensis*; C) Syntenic relationships between scaffolds containing Hox and ParaHox genes in cnidarians supporting gene assignments.”

10) No title is given for reference 74 of the supplementary information.

Response 10:

The title has been added back to the reference.